# Dynamic Incentive-aware Learning: Robust Pricing in Contextual Auctions

**Negin Golrezaei**
Sloan School of Management
Massachusetts Institute of Technology
Cambridge, MA
golrezae@mit.edu

**Adel Javanmard**
Data Sciences and Operations Department
University of Southern California
Los Angeles, CA
ajavanma@usc.edu

**Vahab Mirrokni**
Google Research
New York, NY
mirrokni@google.com

## Abstract

Motivated by pricing in ad exchange markets, we consider the problem of robust learning of reserve prices against strategic buyers in repeated contextual second-price auctions. Buyers' valuations for an item depend on the context that describes the item. However, the seller is not aware of the relationship between the context and buyers' valuations, i.e., buyers' preferences. The seller's goal is to design a learning policy to set reserve prices via observing the past sales data, and her objective is to minimize her regret for revenue, where the regret is computed against a clairvoyant policy that knows buyers' heterogeneous preferences. Given the seller's goal, utility-maximizing buyers have the incentive to bid untruthfully in order to manipulate the seller's learning policy. We propose two learning policies that are robust to such strategic behavior. These policies use the outcomes of the auctions, rather than the submitted bids, to estimate the preferences while controlling the long-term effect of the outcome of each auction on the future reserve prices. The first policy called Contextual Robust Pricing (CORP) is designed for the setting where the market noise distribution is known to the seller and achieves a T-period regret of $O(d \log(Td) \log(T))$, where $d$ is the dimension of the contextual information. The second policy, which is a variant of the first policy, is called Stable CORP (SCORP). This policy is tailored to the setting where the market noise distribution is unknown to the seller and belongs to an ambiguity set. We show that the SCORP policy has a T-period regret of $O(\sqrt{d \log(Td)} \, T^{2/3})$.

## 1 Introduction

In many online marketplaces, both sides of the market have access to rich dynamic contextual information about the products being sold over time. On the buy side, such information can influence the willingness-to-pay of the buyers for the products, potentially in a heterogeneous way. On the sell side, the information can help the seller differentiate the products and set contextual and possibly personalized prices. To do so, the seller needs to learn the impact of this information on buyers' willingness-to-pay. Such contextual learning can be challenging for the seller when there are repeated interactions between the buy and the sell sides. With repeated interactions, the utility-maximizing buyers may have the incentive to act strategically and trick the learning policy of the seller into lowering the prices. Motivated by this, our key research question is as follows: *How can the seller*

*dynamically optimize (personalized) prices in a robust manner, taking into account the strategic behavior of the buyers?*

One of the online marketplaces that faces this problem is online advertising market. In this market, a prevalent approach to sell online ads is via running real-time second-price auctions in which the advertisers can use an abundance of detailed contextual information before deciding what to bid. In this practice, advertisers can target Internet users based on their (heterogeneous) preferences and targeting criteria. Targeting can create a thin and uncompetitive market in which few advertisers show an interest in each auction. In such a thin market, it is crucial for the ad exchanges to effectively optimize the reserve prices in order to boost their revenue. However, learning the optimal reserve prices is rather difficult due to frequent interactions between advertisers and ad exchanges.

Inspired by this environment, we study a model in which a seller runs repeated second-price auctions with reserve over time. In each auction, an item is being sold to at most one of the buyers. The valuation (willingness-to-pay) of each buyer for the item in period $t$, which is his private information, depends on an observable $d$-dimensional contextual information in that period and his preference vector. We focus on an important special case of this contextual-based valuation model in which the buyer's value is a linear function of his preference vector and contextual information plus some random noise term, where the noise models the impact of contexts that are not measured/observed by the seller. The preference vector, which is unknown to the seller, varies across the buyers. Thus, the preference vectors capture heterogeneity in buyers' valuation.

The seller's goal is to design a policy that dynamically learns/optimizes personalized reserve prices. The buyers are fully aware of the learning policy used by the seller and act strategically in order to maximize their (time-discounted) cumulative utility. Dealing with such a strategic population of buyers, the seller aims at extracting as much revenue as the clairvoyant policy that is cognizant of the preference vectors a priori. These vectors determine the relationship between the valuation of the buyers and contextual information. Put differently, the seller would like to minimize her regret where the regret is defined as the difference between the seller's revenue and that under the clairvoyant policy. Note that the clairvoyant policy provides a strong benchmark because the policy posts the optimal personalized reserve prices based on the observed contexts.

As stated earlier, one of the main hurdles in designing a low-regret learning policy in this setting is the frequent interactions between the seller and the buyers. Due to such interactions, the strategic buyers might have the incentive to bid untruthfully. This way, they may sacrifice their short-term utility in order to deceive the seller to post them lower future reserve prices. Thus, while a single shot second-price auction is a truthful mechanism, repeated second-price auctions in which the seller aims at dynamically learning optimal reserve prices of strategic and utility-maximizing buyers may not be truthful. The untruthful bidding behavior of the buyers makes it hard for the seller to learn the optimal reserve prices, and this, in turn, can lead to her revenue loss. This highlights the necessity to design a robust learning policy that reduces buyers' incentive to follow untruthful strategy. Beside this hurdle, the availability of the dynamic contextual information requires the seller to change the reserve prices dynamically over time based on the contextual information. To do so, the seller needs to learn how buyers react to such information and based on the reactions, posts (dynamic) personalized reserve prices.

We consider setting where the seller (firm) is more patient than the buyer. We formalize it by considering time-discounted utility for the buyers. This is motivated by various applications. For example, in online advertisement markets, the advertisers (buyers) who retarget Internet users prefer showing their ads to the users who visited their website sooner rather than later. In this paper, we propose two learning policies. The first policy, that we call Contextual Robust Pricing (CORP), is tailored to a setting where the distribution of the noise term in buyers' valuation is known to the seller. We will refer to this noise as market or valuation noise. Our CORP policy gets the cumulative T-period regret of order $O(d \log(Td) \log(T))$, where the regret is computed against the clairvoyant policy that knows the preference vectors as well as the market noise distribution. The second policy, called Stable CORP (SCORP), is a variant of the first policy. This policy lends itself to a setting where the distribution of the market noise is unknown and belongs to an ambiguity set. In this setting, the seller does not have the intention of learning the market noise distribution. She instead would like to design a learning policy that is robust to the uncertainty in the noise distribution. SCORP achieves the T-period regret of order $O(\sqrt{d \log(Td)} \, T^{2/3})$. Here, we highlight two important aspects of these policies. First, they have an episodic structure and update the estimate of preference vectors only at

the beginning of each episode. Such design make the policy robust by restricting the future effect of the submitted bids. Specifically, bids in an episode are not used in choosing the reserve prices until the beginning of the next episode. Therefore, there is always a delay until a buyer observes the effect of a bid on reserves. Then, considering the fact that buyers are impatient and discount the future, they are less incentivized to bid untruthfully. The second important aspect of the policies that ensure robustness is their estimation method of the buyer's preference vectors. Rather than using the submitted bids to estimate the preference vectors, the policy simply uses the *outcome of the auctions*. Because of this feature of the policy, bidding untruthfully does not always result in lower reserve prices; Instead, it can impact the future reserve prices of a buyer only when it leads to changing the outcome of an auction, i.e., when a buyer loses an auction due to underbidding or a buyer wins an auction due to overbidding.

## 2 Related Work

There is a growing body of research on dynamic pricing with learning. Of necessity, we do not provide a complete set of references, and instead refer the reader to [12] for an in-depth survey on this area. In the following we discuss the literature that is closely related to our setting. Recently, several works considered the problem of dynamic pricing in a contextual setting, with non-strategic buyers. [10] studied this problem when the demand function follows the logit model and proposed an ML-based learning algorithm. [24, 11], and [25] proposed a learning algorithm based on the binary search method when the demand function is linear and deterministic. In their models, buyers have homogenous preference vectors and are non-strategic. Hence, the problem reduces to a single buyer setting, where the buyer acts myopically, i.e., the buyer does not consider the impact of the current actions on the future prices. There is also a new line of literature that studied dynamic pricing with demand learning when the contextual information is high dimensional (but sparse); see [19, 6]. Similar problems have been investigated in [18] (assuming varying coefficient valuation models) and [20] (considering a setting where multiple products are offered at each round).

As mentioned earlier, in our setting, the seller repeatedly interacts with a small number of strategic and heterogeneous buyers. We note that [13] presented empirical evidence that showed buyers in online advertising markets act strategically. The work [1, 29, 22] examined the problem of dynamic pricing with strategic buyers in a non-contextual environment. In [1, 29], the seller repeatedly interacts with a single strategic buyer via a posted-price mechanism. Similar to our setting, the seller is more patient than the buyer in a sense that the buyer discounts his future utility. [1] showed that no learning algorithm can obtain a sub-linear regret when the buyer is as patient as the seller. In addition, via designing learning policies, they demonstrated that the seller can get a sub-linear regret bound when the buyer is less patient. [22] studied dynamic pricing when a group of strategic buyers competes with each other in repeated non-contextual second-price auctions. Further, it is assumed that products to be sold are ex-ante identical, and that buyers are homogenous and their valuations are all drawn from a single distribution, which is unknown to the seller. With respect to the homogeneity assumption, we point out that there exists empirical evidence that buyers are indeed heterogeneous [16, 21, 15]. It is not surprising that the heterogeneity in the markets makes the design of selling mechanisms more difficult. In addition, such difficulties get more severe when the seller needs to design dynamic selling mechanisms for a group of strategic buyers that compete with each other repeatedly.

Recently, [26] studied a similar problem in a static non-contextual setting with strategic buyers. Assuming that the market power of each buyer is negligible, they design a mechanism that incentivizes the buyers to be truthful in the first place, by using techniques from differential privacy [28]. Closer to the spirit of this paper, [2] studies the problem of pricing inventory in a repeated posted-price auction. The authors propose a pricing algorithm whose regret is in the order of $O(\sqrt{\log T}\, T^{2/3})$ in a contextual setting, against a strategic buyer. [1] We point out that our regret result improves upon [2] in the following directions: $(i)$ We allow for market noise in our model, whereas [2] considers noiseless setting which posits that buyer's valuation is given as a linear function of features. By adding the noise component, we make the model richer. When the noise distribution is known, our CORP policy obtains a T-period regret of $O(d\log(Td)\log(T))$. In addition, when the noise distribution is unknown, our SCORP policy, which is *doubly* robust against strategic buyers and the uncertainty in the noise distribution, obtains a T-period regret of $O(\sqrt{d\log(Td)}\, T^{2/3})$. $(ii)$ We consider a market of strategic buyers who participate in a second-price auction at each round, while [2], motivated by targeting in online advertising, considers a single buyer case. Note that in case of a single buyer, there

is no notion of bid, as the buyer only needs to decide if he is willing to get the item at the posted price. By contrast, in a market of buyers, each submitted bid of a buyer can potentially affect the utility of that buyer (instant and long-term utility), other buyers' utilities and the seller's revenue.[2]

## 3 Model

Before we describe the model, we adopt some notation that will be used throughout the paper. For an integer $a$, we write $[a] = \{1, 2, \ldots, a\}$. In addition, for a vector $v \in \mathbb{R}^d$, we denote its $j^{\text{th}}$ coordinates by $v_j$, for $j \in [d]$, and indicate its $\ell_2$ norm by $\|v\|$. For two vectors $v, u \in \mathbb{R}^d$, $\langle u, v \rangle = \sum_{j=1}^{d} u_j v_j$ represents their inner product.

We consider a firm who runs repeated second-price auctions with *personalized* reserve over a finite time horizon with length $T$. In each period $t \geq 1$, the firm would like to sell an item to one of $N$ buyers. The item in period $t$ is represented by an observable feature (context) vector denoted by $x_t \in \mathbb{R}^d$. We assume that the features are drawn independently from a fixed distribution $\mathcal{D}$, with a bounded support $\mathcal{X} \subseteq \mathbb{R}^d$. Note that the length of the time horizon $T$ and distribution $\mathcal{D}$ are unknown to the firm. For the sake of normalization and without loss of generality, we assume that $\|x_t\| \leq 1$, and hence take $\mathcal{X} = \{x \in \mathbb{R}^d : \|x\| \leq 1\}$. We let $\Sigma_x = \mathbb{E}[x_t x_t^\mathsf{T}]$ be the second moment matrix of distribution $\mathcal{D}$, and assume that $\Sigma_x$ is a positive definite matrix, where $\Sigma_x$ is unknown to the firm.

For buyers' valuations, we consider a feature-based model that captures heterogeneity among the buyers. In the following, we discuss the specifics of the valuation model. Valuation of buyer $i \in [N]$ for an item in period $t \geq 1$ depends on the feature vector $x_t$ and period $t$ and is denoted by $v_{it}(x_t)$. We assume that $v_{it}(x_t)$ is a linear function of a preference vector $\beta_i$ and the feature vector $x_t$:

$$v_{it}(x_t) = \langle x_t, \beta_i \rangle + z_{it} \quad i \in [N], \ t \geq 1. \tag{1}$$

Whenever it is clear from the context, we may remove the dependency of valuation $v_{it}(x_t)$ on the feature vector $x_t$ and denote it by $v_{it}$. Here, $\beta_i \in \mathbb{R}^d$ represents the buyer $i$'s preference vector, and for the sake of normalization, we assume that $\|\beta_i\| \leq B_p$, $i \in [N]$, where $B_p$ is a constant. The terms $z_{it}$'s, $i \in [N]$, $t \geq 1$, which are independent of the feature vector $x_t$, are idiosyncratic shocks and are referred to as noise. The noise terms are drawn independently and identically from a mean zero distribution $F : [-B_n, B_n] \to [0, 1]$ with density $f : [-B_n, B_n] \to \mathbb{R}^+$, where $B_n$ is a constant.[3] We assume that the firm knows the distribution of the noise $F$. We relax this assumption later in Section 5. Note that the valuation of buyer $i$, $v_{it}$, is not known to the firm, as the preference vector $\beta_i$ and realization of the noise $z_{it}$ are not observable to her. In addition, by our normalization, $v_{it}(x_t) \leq B$, with $B = B_p + B_n$.

We make the following assumption on distribution of the noise $F$.

**Assumption 3.1** (Log-concavity). *$F(z)$ and $1 - F(z)$ are log-concave in $z \in [-B_n, B_n]$.*

Assumption 3.1, which is prevalent in the economics literature [5], holds by several common probability distributions including uniform, and (truncated) Laplace, exponential, and logistic distributions. A few remarks are in order regarding Assumption 3.1. If distribution F is log-concave and its density $f$ is symmetric, i.e., $f(z) = f(-z)$, then $1 - F(z) = F(-z)$ is also log-concave. Moreover, if density $f$ is log-concave, the distribution $F$ is also log-concave [8]. This implies that Assumption 3.1 is satisfied when density $f$ is symmetric and log-concave. We also point out that if a distribution has a monotone hazard rate (MHR), i.e., $\frac{1-F(z)}{f(z)}$ is decreasing in $z$, then $1 - F(z)$ is log-concave. This point, in turn, shows that all MHR and symmetric distributions satisfy Assumption 3.1.

We next describe the repeated second-price auctions and discuss the firm's problem. The goal of the firm is to maximize the cumulative expected revenue in repeated second-price auctions. The firm tries to achieve this by choosing reserves in a *dynamic* and *personalized* manner.

### 3.1 Second-price Auctions with Dynamic Personalized Reserves

Before defining a second-price auction, we need to establish some notation. For buyer $i \in [N]$ and period $t \geq 1$, we let $p_{it}$ be the payment from buyer $i$ in period $t$. Further, let $q_{it}$ be the allocation

variable: $q_{it} = 1$ if the item in period $t$ is allocated to buyer $i$ and is zero otherwise. We also let $b_{it}$ be the bid submitted by buyer $i$ and $r_{it}$ be the reserve price posted by the firm for buyer $i$ in period $t$. We define $\mathbf{b}_t = (b_{1t}, \dots, b_{Nt})$ and $\mathbf{r} = (r_{1t}, \dots, r_{Nt})$ as the vectors of bids and reserves in period $t$, respectively. Moreover, we denote by $H_\tau$ the history set observed by the firm up to period $\tau$. This set includes buyers' bids and reserve prices for all $t < \tau$:

$$H_\tau = \{(\mathbf{r}_1, \mathbf{b}_1), \dots, (\mathbf{r}_{\tau-1}, \mathbf{b}_{\tau-1})\}. \tag{2}$$

Below, we explain the details of the second-price auction with reserve. In period $t \geq 1$,

- The firm observes the feature vector $x_t \sim \mathcal{D}$. In addition, each buyer $i \in [N]$ learns his valuation $v_{it}$, defined in Eq. (1).
- For each buyer, the firm computes reserve price $r_{it}$, as a function of history set $H_t$.
- Each buyer $i \in [N]$ submits a bid of $b_{it}$.
- Let $i^\star = \arg\max_{i \in [N]}\{b_{it}\}$. If $b_{i^\star t} \geq r_{i^\star t}$, then the item is allocated to buyer $i^\star$, and we have $q_{i^\star t} = 1$. In case of tie, the item is allocated uniformly at random to one of the buyers among those with the highest bid. For all buyers who do not get the item, we have $q_{it} = 0$.
- For each buyer $i$, if he gets the item ($q_{it} = 1$), then he pays $p_{it} = \max\{r_{it}, \max_{j \neq i}\{b_{jt}\}\}$. Otherwise, $p_{it} = 0$.

To lighten the notation, we henceforth use the following shorthands. For each period $t$, we let $b_t^+$ and $b_t^-$ respectively denote the highest and second highest bids. Likewise, we define $v_t^+$ and $v_t^-$ as the highest and second highest valuations in period $t$. We also let $r_t^+$ be the reserve price of the buyer with the highest bid. Therefore, $b_{i^\star t} = b_t^+$, $r_{i^\star t} = r_t^+$, and the firm receives a payment of $\max\{r_t^+, b_t^-\}$ if the item gets allocated and zero otherwise. We assume that for all periods $t$, $b_t^+ \leq M$ for some constant $M > 0$. In words, buyers submit bounded bids.

The firm's decision in any period $t \geq 1$ is to find optimal reserve price $r_{it}$, $i \in [N]$, and her objective is to maximize her (cumulative) expected revenue. Note that revenue of the firm is the total payment she collects from the buyers over the length of the time horizon. Let

$$\mathsf{rev}_t = \mathbb{E}\Big[\sum_{i \in [N]} p_{it} q_{it}\Big] = \mathbb{E}\big[\max\{b_t^-, r_t^+\}\mathbb{I}(b_t^+ \geq r_t^+)\big] \tag{3}$$

be the expected revenue of the firm in period $t \geq 1$, where the expectation is w.r.t. to the noise distribution $F$, feature distribution $\mathcal{D}$, and any randomness in the bidding strategy of buyers and learning policy used by the firm. Then, the total revenue of the firm is given by $\sum_{t=1}^{T} \mathsf{rev}_t$. Maximizing the firm's revenue is equivalent to minimizing her regret where the regret is defined as the difference between the firms' revenue and the maximum expected revenue that the firm could earn if she knew the preference vectors $\{\beta_i\}_{i \in [N]}$. In the next section, we will formally define the firm's regret.

## 3.2 Benchmark and Firm's Regret

When the preference vectors and noise distribution $F$ are known, to set the optimal reserves $r_{it}$, the benchmark policy does not need any knowledge from the history set $H_t$. Thus, with the knowledge of the preference vectors, all buyers are incentivized to bid truthfully against the benchmark policy. This is the case because single-shot second-price auctions are strategy proof [30]. We next characterize the benchmark policy. Let $r_{it}^\star$ be the reserve of buyer $i$ in period $t$ posted by the benchmark policy and following our convention, we denote by $r_t^{\star+}$ the reserve price of the buyer with the highest bid.

**Proposition 3.2** (Benchmark). *If the firm knows the preference vectors $\{\beta_i\}_{i \in [N]}$, then the optimal reserve price of buyer $i \in [N]$ for a feature vector $x \in \mathcal{X}$ is given by*

$$r_i^\star(x) = \arg\max_y \{y(1 - F(y - \langle x, \beta_i \rangle))\} \quad i \in [N], \; x \in \mathcal{X}, \tag{4}$$

*and hence $r_{it}^\star = r_i^\star(x_t)$. In addition, in any period $t \geq 1$, the benchmark expected revenue is given by*

$$\mathsf{rev}_t^\star = \mathbb{E}\big[\max\{v_t^-, r_t^{\star+}\}\mathbb{I}(v_t^+ \geq r_t^{\star+})\big], \tag{5}$$

*where expectation is w.r.t. to the noise distribution $F$ and the feature distribution $\mathcal{D}$.*

We refer to Appendix E for the proof of Proposition 3.2. We remark that the benchmark revenue $\text{rev}_t^\star$ is measured against truthful buyers, while the firm's revenue under our policy is measured against strategic buyers who may not necessarily follow the truthful strategy. Observe that the optimal reserve price of buyer $i$ in period $t$, denoted by $r_{it}^\star$, solves the following optimization problem

$$r_{it}^\star \;=\; \arg\max_y \; \{y \cdot \mathbb{P}\left(v_{it}(x_t) \geq y\right)\} \;=\; \arg\max_y \left\{y \cdot \mathbb{P}\left(\langle x_t, \beta_i \rangle + z_{it} \geq y\right)\right\}.$$

This shows that the optimal reserve price of buyer $i$ does not depend on the number of buyers participating in the auction or their preference vectors. In other words, in (lazy) second-price auctions, when the preference vectors are known to the firm, the problem of optimizing reserve prices can be decoupled. Because of this, the benchmark, defined in Proposition 3.2, has a simple structure: For any feature vector $x \in \mathcal{X}$, the optimal reserve price of buyer $i$, $r_i^\star(x)$, only depends on $\beta_i$ and feature $x$, and is independent of $\beta_j$, $j \neq i$.

Having defined the benchmark, we are now ready to formally define the regret of a firm's policy $\pi$. Consider a policy $\pi$ that posts a vector of reserve prices $\mathbf{r}_t^\pi = (r_{1t}^\pi, \ldots, r_{Nt}^\pi)$, as a function of history set $H_t$ observed by the firm. Suppose that the buyers submit bids of $\mathbf{b}_t = (b_{1t}, \ldots, b_{Nt})$, $t \geq 1$, where $\mathbf{b}_t$ may not be equal to the vector of valuations $\mathbf{v}_t = (v_{1t}, \ldots, v_{Nt})$. The submitted bid of buyer $i$, $b_{it}$, can depend on the learning policy used by the firm, context $x_t$, his valuation $v_{it}$, and history $H_{it}$, where

$$H_{it} = \{(v_{i1}, b_{i1}, q_{i1}, p_{i1}), \ldots, (v_{i(t-1)}, b_{i(t-1)}, q_{i(t-1)}, p_{i(t-1)})\}.$$

Recalling our notation, we write $r_t^{\pi+}$ to denote the reserve price, set by policy $\pi$, of the buyer with the highest bid in period $t$. Then, the expected revenue of the firm under policy $\pi$ in period $t$ reads as

$$\text{rev}_t^\pi \;=\; \mathbb{E}\left[\max\{b_t^-, r_t^{\pi+}\}\mathbb{I}(b_t^+ \geq r_t^{\pi+})\right], \tag{6}$$

where expectation is w.r.t. to the noise distribution $F$, feature distribution $\mathcal{D}$, and any randomness in bidding strategy of the buyers. Then, the worst-case cumulative regret of policy $\pi$ is defined by

$$\text{Reg}^\pi(T) = \max\left\{\sum_{t=1}^T (\text{rev}_t^\star - \text{rev}_t^\pi) : \|\beta_i\| \leq B_p, \text{ for } i \in [N], \text{supp}(\mathcal{D}) \subseteq \mathcal{X}\right\}. \tag{7}$$

Note that the regret of the policy $\pi$ is not a function of the feature distribution $\mathcal{D}$ and the feature vectors $\{\beta_i\}_{i \in [N]}$. That is, we compute the regret of the policy $\pi$ against the worst feature distribution $\mathcal{D}$ and preference vectors $\{\beta_i\}_{i \in [N]}$. In the next section, we discuss buyers' bidding behavior.

### 3.3 Utility-maximizing Buyers

We assume that each buyer $i \in [N]$ is risk neutral and aims at maximizing his (time-discounted) cumulative expected utility. The utility of buyer $i$ in period $t \geq 1$ with valuation $v_{it}$ is given by $u_{it} = v_{it}q_{it} - p_{it}$. Note that through the allocation variables $q_{it}$, utility $u_{it}$, depends on the submitted bids of all the buyers, $\mathbf{b}_t$, and their reserve price $\mathbf{r}_t$ used by the firm.

Each buyer $i$ would like to maximize his time-discounted cumulative utility, which is defined as $U_i = \sum_{t=1}^\infty \gamma^t \mathbb{E}[u_{it}]$, where $\gamma \in (0, 1)$ is a discount factor. The discount factor highlights the fact that the firm is more patient than the buyers. For instance, in online advertising markets, advertisers are willing to show their ads to the users who just visited their websites.[4] As another example, in cloud computing markets, the consumers would like to access enough capacity whenever they need it [7]. We note that [1] showed that it is impossible to get a sub-linear regret when buyers are utility-maximizer and do not discount their future utilities.

*All buyers fully know the learning policy that the firm is using to set the reserves.*[5] Armed with this knowledge, buyers can potentially increase their future utility they earn via bidding untruthfully. Particularly, a buyer can underbid (shade) his bid by submitting bid $b_{it} < v_{it}$, or he can overbid by submitting bid $b_{it} > v_{it}$. Both shading and overbidding can potentially impact the firms' estimate of preference vectors of the buyers and this, in turn, can hurt the firms' revenue. However, shading can lead to a utility loss in the current period, as by shading, the buyer may lose an auction that he would have won by bidding truthfully. Similarly, overbidding can result in a utility loss in the current period, as by overbidding the buyer might end up paying more than his valuation.

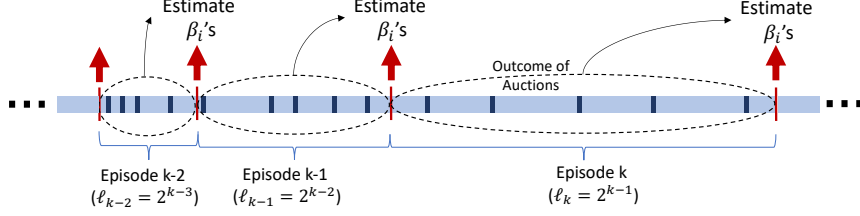

Figure 1: Schematic representation of the CORP policy. The dark blue rectangles show the random exploration periods.

## 4  CORP: A Contextual Robust Pricing Policy

In this section, we present our learning policy. The description of the policy is provided in Table 1. For reader's convenience, we also provide a schematic representation of CORP in Figure 1. The policy works in an episodic manner. It tries to learn the preference vectors by using Maximum Likelihood Estimation (MLE) and meanwhile sets the reserve prices based on its current estimates of the preference vectors. Episodes are indexed by $k = 1, 2, \ldots$, where the length of each episode, denoted by $\ell_k$, is given by $2^{k-1}$. Thus, episode $k$ starts in period $\ell_k = 2^{k-1}$ and ends in period $\ell_{k+1} - 1 = 2^k - 1$. Note that the length of episodes increases exponentially with $k$. Throughout, we use notation $E_k$ to refer to periods in episode $k$, i.e., $E_k \equiv \{\ell_k, \ldots, \ell_{k+1} - 1\}$.

At the beginning of each episode $k$, we estimate the preference vectors of the buyers using the outcome of the auctions ($q_{it}$'s) in the pervious episode, i.e., episode $k - 1$, and we do not change our estimates during episode $k$. Let $\widehat{\beta}_{ik}$ be the estimated preference vector of buyer $i$ at the beginning of episode $k$. Then, $\widehat{\beta}_{ik}$ solves the following optimization problem:

$$\widehat{\beta}_{ik} = \underset{\|\beta\| \leq B_p}{\arg\min} \; \mathcal{L}_{ik}(\beta), \;\; i \in [N], \tag{8}$$

where

$$\mathcal{L}_{ik}(\beta) = -\frac{1}{\ell_{k-1}} \sum_{t \in E_{k-1}} \big\{ q_{it} \log \big( (1 - F(\max\{b^+_{-it}, r_{it}\} - \langle x_t, \beta \rangle)) \big)$$
$$+ (1 - q_{it}) \log \big( F(\max\{b^+_{-it}, r_{it}\} - \langle x_t, \beta \rangle) \big) \big\} \tag{9}$$

is the negative of the log-likelihood function. Here, $b^+_{-it}$ refers to the maximum bids of buyers other than buyer $i$, in period $t$; that is, $b^+_{-it} = \max_{j \neq i} b_{jt}$. Then, buyer $i$ wins the auction in period $t$ if and only if $b_{it} > \max\{b^+_{-it}, r_{it}\}$. Similarly, we define $v^+_{-it} = \max_{j \neq i} v_{jt}$. Note that $F(\max\{b^+_{-it}, r_{it}\} - \langle x_t, \beta \rangle)$ is the probability of event $\langle x_t, \beta \rangle + z_{it} \leq \max\{b^+_{-it}, r_{it}\}$, which is the probability that buyer $i$ does not win the item at time $t$, upon bidding truthfully. The log-likelihood function $\mathcal{L}_{ik}(\beta)$ is computed after running the auctions in all the periods of episode $E_{k-1}$. Therefore, the firm has access to the required knowledge to compute the log-likelihood function $\mathcal{L}_{ik}(\beta)$. Specifically, by the time the firm computes $\mathcal{L}_{ik}(\beta)$, she has access to the submitted bids of the buyers in periods $t \in E_{k-1}$ as well as the reserve prices used in these periods. After estimating the preference vectors at the beginning of each episode $k$, the policy proceeds to use its estimation to set reserve prices. In particular, inspired by Proposition 3.2, the reserve price in period $t \in E_k$, $r_{it}$, solves (11).

We now discuss some of the important features of our policy.

($i$) In each episode $k$, every period $t$ is assigned to exploitation with probability $1 - 1/\ell_k$, and is assigned to exploration with probability $1/\ell_k$. In the exploration periods, the firm chooses one of the buyers at random and allocates the item to him if his submitted bid is above a reserve price $r \sim \mathsf{uniform}(0, B)$ where $\mathsf{uniform}(0, B)$ is the uniform distribution in the range $[0, B]$. In exploitation periods, the firm exploits her current estimate of the preference vectors to set the reserve prices where the estimates are obtained by applying the MLE method to the outcomes of auctions in episode $k - 1$. The main purpose of setting reserve prices randomly in the exploration periods is to motivate the buyers to be truthful. Note that the buyer does not know if in a given period t, the prices are set randomly. Thus, if he underbids in such a period, with a positive probability, he loses the opportunity to obtain a positive utility.

<div style="border:1px solid black; padding:10px">

**CORP: A Contextual Robust Pricing**

Initialization: For any $k \in \mathbb{Z}^+$, let $\ell_k = 2^{k-1}$ and $E_k = \{\ell_k, \ldots, \ell_{k+1} - 1\}$. Moreover, we let $r_{i1} = 0$ and $\widehat{\beta}_{i1} = 0$ for any $i \in [N]$.

Updating Preference Vectors: At the start of each episode $k = 1, 2, \ldots$, i.e, at the beginning of period $t = \ell_k$, estimate the preference vectors, denoted by $\{\widehat{\beta}_{ik}\}_{i \in [N]}$, as follows

$$\widehat{\beta}_{ik} = \arg\min_{\|\beta\| \le B_p} \mathcal{L}_{ik}(\beta), \ \ i \in [N], \tag{10}$$

where $\mathcal{L}_{ik}(\beta)$ is defined in Eq. (9).

Setting Reserves: In each episode $k = 1, 2, \ldots$, and for any period $t$ in this episode, i.e., $t \in E_k$,

- *Exploration Phase:* With probability $\frac{1}{\ell_k}$, choose one of the $N$ buyers uniformly at random and offer him the item at price of $r \sim \mathsf{uniform}(0, B)$, where $\mathsf{uniform}(0, B)$ is the uniform distribution in the range $[0, B]$. For other buyers, set their reserve prices to $\infty$.
- *Exploitation Phase:* With probability $1 - \frac{1}{\ell_k}$, observe the feature vector $x_t$ and set the reserve of each buyer $i \in [N]$ to

$$r_{it} = \arg\max_y \left\{ y\big(1 - F(y - \langle x_t, \widehat{\beta}_{ik} \rangle)\big) \right\}. \tag{11}$$

</div>

**Table 1:** CORP Policy

($ii$) We highlight that CORP policy does not use the submitted bids in estimating the preference vectors: It only uses the outcomes of the auctions, i.e., $q_{it}$'s, to estimate these vectors; see the definition of the log-likelihood function in Equation (9). This makes the estimation procedure of the policy robust to untruthful bidding behavior of the buyers, as untruthful bidding may not necessarily lead to a different outcome. In addition, due to this feature of the learning policy, the buyers are incentivized to bid truthfully unless they are interested in changing the outcome of the auction at the expense of losing their current utility.

($iii$) Other important factors that makes the CORP policy robust is its episodic structure and impatience of buyers. In the CORP policy, submitted bids in episode $k$ are not used in setting reserve prices until the beginning of episode $(k + 1)$. Therefore, there is always a delay until buyers observe the effect of a bid on their reserves. Then, since buyers are impatient and maximize their discounted cumulative utility, they have less incentive to bid untruthfully. This is a salient property of the CORP policy that bounds the perpetual effect of each bid and, as we will see in the analysis, leads to robustness of the learning policy to the strategic behavior of buyers.

**Theorem 4.1** (Regret Bound: Known Market Noise Distribution). *Suppose that Assumption 3.1 holds and the firm knows the market noise distribution $F$. Then, the T-period worst-case regret of the CORP policy is at most $O(d\log(Td)\log(T))$, where the regret is computed against the benchmark, defined in Proposition 3.2.*

In Appendix A we give a proof sketch of Theorem 4.1 and refer to Appendix C for a detailed proof.

## 5 SCORP: Stable CORP Policy

The CORP policy is assumed to know the market noise distribution $F$. Nevertheless, in practice, it may very well be the case that distribution $F$ is unknown or cannot be well approximated (e.g., it changes over time). To address this problem, we propose a variant of the CORP policy, called Stable Contextual Robust Pricing (SCORP), which is robust against the lack of a precise knowledge of $F$. Specifically, we consider an *ambiguity* set $\mathcal{F}$ of possible probability distributions for the market noise and propose a policy that works well for every distribution in the ambiguity set.

Due to space constraint, we briefly explain SCORP here and refer to Appendix B for more details and a formal description of the policy. Similar to the COPR policy, SCORP has an episodic theme, with the length of episodes growing exponentially. As before, we denote the set of periods in episode $k$ by $E_k$, i.e., $E_k = \{\ell_k, \ldots, \ell_{k+1} - 1\}$, with $\ell_k = 2^{k-1}$. However, instead of having randomized

exploration, each episode $k$ starts with a pure exploration phase of length $\lceil \ell_k^{2/3} \rceil$. We use notation $I_k$ to refer to periods in the pure exploration phase of episode $k$, i.e., $I_k \equiv \{\ell_k, \ldots, \ell_k + \lceil \ell_k^{2/3} \rceil\}$. During each period in $I_k$, we choose one of the $N$ buyers uniformly at random and offer him the item at price of $r \sim \text{uniform}(0, B)$. For other buyers, we set their reserve prices to $\infty$. In the remaining periods of the episode (i.e., $E_k \backslash I_k$), we offer the reserve prices based on the current estimates of the preference vectors which are obtained by applying the least-square estimator to the outcomes of auctions in the pure exploration phase, $I_k$. This is the exploitation phase as we set reserves based on our best guess of the preference vectors. In the least-square estimator, SCORP uses the outcome of the auctions, *not* the submitted bids, which makes SCORP robust to the strategic buyers. In addition, in the exploitation phase, SCORP chooses reserve prices in a way to maximize the worst case revenue, over the ambiguity set $\mathcal{F}$, based on the current estimate of the preference vectors. In this sense, SCORP is robust also against the uncertainty in the noise distribution. Thus, SCORP is indeed doubly robust. In Theorem B.3 in Appendix B, we show that SCORP achieves the T-period worst-case regret of $O(\sqrt{d \log(Td)} \, T^{2/3})$.

# 6 Extension to nonlinear models

Although the paper focuses on linear valuation models, it is straightforward to generalize our analysis to some of the nonlinear valuation models. Specifically, consider model

$$ v_{it}(x_t) = \psi(\langle \phi(x_t), \beta_i \rangle + z_{it}) \quad i \in [N], \quad t \geq 1 \,, $$

where $\phi : \mathbb{R}^d \mapsto \mathbb{R}^d$ is a mapping and $\psi : \mathbb{R} \mapsto \mathbb{R}$ is an increasing function. Then by the change of variable $\tilde{v}_{it} = \psi^{-1}(v_{it})$, $\tilde{x}_t = \phi(x_t)$, we arrive at the relation $\tilde{v}_{it} = \langle \tilde{x}_t, \beta_i \rangle + z_{it}$. By modifying the CORP policy for this relation, we can get a policy that also achieve logarithmic regret for these nonlinear settings. Some examples include: log-log model, semi-log model and logistic model. We refer to the long version of the paper [14] for further discussion on this matter.

While these models have been popular for some applications (the first two in hedonic pricing and the last in click-through-rate prediction), it is still an interesting direction to consider nonparametric models (similar to [27] and [9] as examples) but it is beyond the scope of current paper.

# 7 Conclusion

Motivated by online marketplaces with highly differentiated products, we formulated a dynamic pricing problem in the contextual setting. In this problem, a firm runs repeated second-price auctions with reserve and the item to be sold in each period is described by a context (feature) vector. In our model, contextual information of an item influences buyers' valuations of that item in a heterogeneous way, via buyers' preference vectors. Due to the repeated interaction of buyers with the firm, buyers have the incentive to game the firm's policy by bidding untruthfully. We proposed two pricing policies to set the reserve prices of buyers. These policies aim at learning the preference vectors of buyers in a robust way against strategic buyers and meanwhile maximize the firm's collected revenue.

The main insight behind the robustness property of our approach is that by an episodic design, we limit the long-term effect of each bid on the firm's estimates of the preference vectors. Further, instead of using the bids (data) we use only the outcomes of auctions (censored data) in estimating preference vectors. Interestingly, we show that using this censored data does not hamper the learning rate while bringing in robustness property. As the granularity of real-time data increases at an unprecedented rate, we believe the ideas of this work can serve as a starting point for other complex dynamic contextual learning and decision making problems.

## Acknowledgement

A. Javanmard was supported in part by an Outlier Research in Business (iORB) grant from the USC Marshall School of Business, a Google Faculty Research Award and the NSF CAREER Award DMS-1844481. A. Javanmard would also like to acknowledge the financial support of the Office of the Provost at the University of Southern California through the Zumberge Fund Individual Grant Program.

## Footnotes

[1]Dependency on $d$ is hidden in the big-$O$ notation.

[2] Section 5 in [2] considers an extension to the multiple buyers case but assumes that the highest valuation in each period $t$ can be written as $\langle x_t, \beta \rangle$ for a fixed parameter vector $\beta$, and product feature (context) $x_t$, which we find to be a strong assumption.

[3] The noise aims at capturing features that are not observed/measured by the firm.

[4]Such a practice is known as retargeting [2, 15].

[5]This assumption is inspired by the literature on the behavior-based pricing where it is shown that the firm can earn more revenue by committing to a pricing strategy [17, 31]. See also [3, 4] for a similar insight.

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
