[Supplementary Material]

# A    Proof Sketch of Theorem 4.1

To bound the regret of our policy, we note that untruthful bidding has two undesirable effects for the firm. First of all, both overbidding and shading increase the estimation errors of the preference vectors and this, consequently, introduce errors in the reserve prices set by the firm. Second of all, bidding untruthfully can lower the second highest submitted bid as well as the reserve price of the winner, and this reduces the firm's revenue. By this observation, we divide the policy's regret into two parts, where each part captures the negative consequences of one of the aforementioned effects.

To bound the regret associated with the first effect, in Proposition C.1, we determine to which extent buyers' "lies" impact the estimation errors of the preference vectors and the reserve prices. We say a buyer lies when his untruthful bid changes the outcome of the auction, $q_{it}$, for this buyer relative to the truthful bidding; see Equation (17) for a formal definition. We note that by changing the outcome of the auction in a period $t \in E_k$, the buyer can game the firm's learning policy and hence can decrease his reserve prices in episode $k + 1$. Having established the impact of lies, we then show that the number of times that a buyer "lies" in each episode is logarithmic in the length of the episode; see Proposition C.2.

This bound is derived using the fact that buyers are utility-maximizer and discount the future. To get this bound, we compare the long-term excess utility obtained from a lie with the instant utility loss that it causes. In particular, we derive a lower bound on the utility loss of the buyers in episode $k$ by focusing on the random exploration periods. Note that buyers are not aware whether a period is an exploration period. Thus, with a positive probability, any untruthful bid leads to a utility loss. We further derive an upper bound on the (future) utility gain of the untruthful bidding in episode $k$. Our upper bound is the total discounted utility that any buyer can hope to achieve in the next episodes. Thus, the bound includes potential future utility gains that a buyer can enjoy by manipulating other buyers' strategy and their reserve prices. Then, by arguing that for any utility-maximizer buyer, the upper bound on the utility gain should be greater than or equal to the lower bound on the utility loss, we bound the number of lies of the buyer. By characterizing the impact of lies in Proposition C.1 and bounding the number of lies in Proposition C.2, we are able to bound the regret associated with the first effect, namely the gap between the posted reserves and the optimal ones.

To bound the regret associated with the second effect, we quantify the impact of bidding untruthfully on the second highest bids and reserve of the winner; see Lemma C.5. When buyers bid untruthfully, the second highest bid may decrease. Further, the winner of the auction can change and this, in turn, can lower the reserve price of the winner. Any of these events will hurt the firm's revenue. To quantify this impact, we upper bound the amount of underbidding and overbidding from each buyer using the fact that the buyer is utility-maximizer; see Proposition C.2. To do so, we employ a similar argument that we used to bound the number of lies.

After characterizing the regret due to both effects, we bound the total regret during each episode, and show that the total regret is logarithmic in the length of the episode. The proof is completed by noting that there are $O(\log T)$ episodes up to time $T$, as the length of episodes doubles each time.

# B    SCORP: Stable CORP Policy

As discussed in the paper, the Stable Contextual Robust Pricing (SCORP) is a variant of the CORP policy designed for the setting where the market noise distribution is unknown to the seller. Specifically, we consider an *ambiguity* set $\mathcal{F}$ of possible probability distributions for the market noise and propose a policy that works well for every probability distribution in the ambiguity set.

We make the following assumption on the ambiguity set $\mathcal{F}$. This assumption is analogous to Assumption 3.1.

**Assumption B.1** (Log-concavity of the Ambiguity Set $\mathcal{F}$). *All functions $F \in \mathcal{F}$ are log-concave.*

To be fair, in this case, we compare the regret of our policy against a benchmark policy, called stable, that knows the true preference vectors $\beta_i$ and the ambiguity set $\mathcal{F}$ that includes $F$, but is oblivious to distribution $F$ itself. The stable benchmark is defined as follows:

Figure 2: Schematic representation of the SCORP policy.

**Definition B.2** (Stable Benchmark). *In the stable benchmark, the reserve price of buyer $i \in [N]$ for a feature vector $x \in \mathcal{X}$ is given by*

$$r_i^\star(x) = \arg\max_y \ \min_{F \in \mathcal{F}} \left\{ y\left(1 - F(y - \langle x, \beta_i \rangle)\right) \right\}, \quad i \in [N], \quad x \in \mathcal{X}, \tag{12}$$

*and thus $r_{it}^\star = r_i^\star(x_t)$ in this case.*

The stable benchmark is motivated by our previous benchmark presented in Proposition 3.2. In our previous benchmark, we show that given a distribution $F$ and context $x$, the revenue-maximizing reserve price for buyer $i$ solves $r_i^\star(x) = \arg\max_y \left\{ y\left(1 - F(y - \langle x, \beta_i \rangle)\right) \right\}$. In the stable benchmark, the posted reserve prices have a similar form. However, the reserve prices are chosen in a robust way so that the benchmark performs well despite the uncertainty in the market noise distribution.

We note that the stable benchmark as well as our learning policy that we will present shortly do not aim at learning the distribution of market noise, as this distribution can vary across periods. For instance, in online advertising, the distribution of the noise can depend on many different factors including the time of the day and demographic information of the Internet users. Thus, instead of trying to learn the market noise distribution, we would like to use reserve prices that are robust to the uncertainty in the noise distribution.

We are now ready to present our SCORP policy. This policy is a modified version of the CORP policy. For reader's convenience, we also provide a schematic representation of SCORP in Figure 2. Similar to the COPR policy, SCORP has an episodic theme, with the length of episodes growing exponentially. As before, we denote the set of periods in episode $k$ by $E_k$, i.e., $E_k = \{\ell_k, \ldots, \ell_{k+1} - 1\}$, with $\ell_k = 2^{k-1}$. However, instead of having randomized exploration, each episode $k$ starts with a pure exploration phase of length $\lceil \ell_k^{2/3} \rceil$. Throughout, we use notation $I_k$ to refer to periods in the pure exploration phase of episode $k$, i.e., $I_k \equiv \{\ell_k, \ldots, \ell_k + \lceil \ell_k^{2/3} \rceil\}$. During each period in $I_k$, we choose one of the $N$ buyers uniformly at random and offer him the item at price of $r \sim \text{uniform}(0, B)$. For other buyers, we set their reserve prices to $\infty$. In the remaining periods of the episode (i.e., $E_k \backslash I_k$), we offer the reserve prices based on the current estimates of the preference vectors which are obtained by applying the least-square estimator to the outcomes of auctions in the pure exploration phase, $I_k$; see Equations (13) and (15). This is the exploitation phase as we set reserves based on our best guess of the preference vectors. Note that the least-square estimator, similar to the CORP policy, SCORP uses the outcome of the auctions, *not* the submitted bids, which makes SCORP robust to the strategic buyers. In addition, the choice of reserve prices in the exploitation phase of SCORP makes this policy robust against the uncertainty in the noise distribution. Thus, SCORP is indeed doubly robust. The formal description of SCORP is given in Table 2.

Before we move to a formal description of SCORP, let us pause to build some insight into the design of SCORP. One of the challenges that SCORP is facing is the uncertainty in the market noise distribution. The market noise, on one hand, makes the estimation of preference vectors hard but on the other hand, the randomness in noise provides us with some extent of exploration. When the noise distribution $F$ is known, our CORP policy harnessed this exploration by forming a log-likelihood estimator, and because of that CORP policy assigns fewer periods to pure exploration.

However, as stated earlier, this picture changes when $F$ is unknown as we cannot effectively leverage the exploration provided by noise. Therefore, we adopt a different approach wherein at the beginning of each episode, we do pure exploration by using random prices. The length of pure exploration phases is designed in a way to ensure that we have enough number of samples to update the estimate of the preference vectors at a proper rate.

<div style="border:1px solid black;padding:10px;">

**SCORP: Stable Contextual Robust Pricing Policy**

Initialization: For any $k \in \mathbb{Z}^+$, let $\ell_k = 2^{k-1}$, $E_k = \{\ell_k, \ldots, \ell_{k+1} - 1\}$, and $I_k = \{\ell_k, \ldots, \ell_k + \lceil \ell_k^{2/3} \rceil\}$. Moreover, we let $r_{i1} = 0$ and $\widehat{\beta}_{i1} = 0$ for any $i \in [N]$.

For $k = 1, 2, \ldots$, do the following steps:

Pure Exploration Phase: For $t \in I_k$, choose one of the $N$ buyers uniformly at random and offer him the item at price of $r \sim \mathsf{uniform}(0, B)$. For other buyers, set their reserve prices to $\infty$.

Updating Estimates: At the end of the exploration phase, update the estimate of the preference vectors by applying the least-square estimator to the previous pure exploration phase:

$$\widehat{\beta}_{ik} = \arg\min_{\|\beta\| \le B_p} \tilde{\mathcal{L}}_{ik}(\beta), \; i \in [N], \tag{13}$$

where $\tilde{\mathcal{L}}_{ik}(\beta)$ is given by

$$\tilde{\mathcal{L}}_{ik}(\beta) = \frac{1}{|I_k|} \sum_{t \in I_k} (BN q_{it} - \langle x_t, \beta \rangle)^2. \tag{14}$$

Exploitation Phase: For $t \in E_k \backslash I_k$, observe the feature vector $x_t$ and set the reserve of each buyer $i \in [N]$ to

$$r_{it} = \arg\max_y \; \min_{F \in \mathcal{F}} \left\{ y \big(1 - F(y - \langle x_t, \widehat{\beta}_{ik} \rangle)\big) \right\}. \tag{15}$$

</div>

**Table 2:** SCORP Policy

Having presented our policy, we now highlight few important remarks about the estimation process of the policy. (i) Since the noise distribution is unknown, SCORP employs the least-square estimator rather than the maximum likelihood method, used in CORP; see Equation (14). To apply the least-square estimator, similar to the CORP policy, SCORP uses the outcome of the auctions, *not* the submitted bids.[6] This makes SCORP robust to the strategic behavior of the buyers. (ii) Due to uncertainty in the noise distribution, for estimation, SCORP only utilizes the auction outcomes in the exploration phase where it does price experimentation. This is in contrast to CORP policy where all the auction outcomes in the previous episode are used to estimate the preference vectors. It is worth noting that in our analysis of the regret, we give up on the revenue collected during pure exploration phases and only use the outcomes of auctions in these phases to bound the estimation error of the preference vectors.

So far, we argued SCORP is designed in a way to ensure robustness against strategic buyers. Importantly, we also note that the choice of reserve prices in the exploitation phase of SCORP makes this policy robust against the uncertainty in the noise distribution. Thus, SCORP is indeed doubly robust.

Our next result upper bounds the regret of the SCORP policy.

**Theorem B.3** (Regret Bound: Unknown Market Noise Distribution)**.** *Suppose Assumption B.1 holds, and that the market noise distribution is unknown and belongs to uncertainty set $\mathcal{F}$. Then, the T-period worst-case regret of the SCORP policy is at most $O(\sqrt{d \log(Td)} \, T^{2/3})$, where the regret is computed against the stable benchmark.*

Observe that while the regret of the CORP policy is in the order of $O(d \log(Td) \log(T))$, the regret of SCORP is $O(\sqrt{d \log(Td)} \, T^{2/3})$. The higher regret of SCORP is mostly due to the uncertainty in the noise distribution. Because of this uncertainty, as stated earlier, SCORP cannot make use of the exploratory effect of the noise. Instead, the SCORP policy dedicates $\lceil \ell_k^{2/3} \rceil$ number of periods in each episode $k$ to pure exploration. This implies that the SCORP policy learns preference vectors at a slower rate than the CORP policy. The slower learning rate is the main reason behind the higher regret of SCORP.

The proof of Theorem B.3 is provided in Appendix D.

**Remark B.4.** *SCORP policy provides a very general machinery to design low-regret doubly robust learning policies, against different benchmarks.[7] To make it clear, assume that firm uses a benchmark that posts reserve price of $r_i^*(x) = \rho(\langle x, \beta_i \rangle, \mathcal{F})$ for buyer $i$, under context vector $x \in \mathcal{X}$. Here, $\rho(\langle x, \beta_i \rangle, \mathcal{F}) = \arg\max_y \min_{F \in \mathcal{F}} G(y, \langle x, \beta_i \rangle, F)$, where $G : \mathbb{R} \times \mathbb{R} \times \mathcal{F} \to \mathbb{R}$. Then, as long as $\rho(\langle x, \beta_i \rangle, \mathcal{F})$ is Lipschitz in its first argument, we can design a low-regret doubly robust learning policies against this benchmark by only changing the exploitation phase of the SCORP policy. Particularly, in a period $t$ in the exploitation phase of episode $k$, we set $r_{it} = \rho(\langle x, \widehat{\beta}_{ik} \rangle, \mathcal{F})$; see Equation (15) for comparison.*

## C Proof of Theorem 4.1

The regret of the CORP policy is the sum of its regret across all episodes. Thus, in the following, we compute the regret incurred during an episode $k > 1$. (The regret of episode 1 that has a length of 1 is a constant.)

We start with a definition. Let

$$l_F = \inf_{|x| \leq B_n} \left\{ \min \left\{ -\log'' F(x), -\log''(1 - F(x)) \right\} \right\}, \tag{16}$$

where $\log'' F(x) = \frac{d^2}{dx^2}(\log(F(x)))$ and $\log''(1 - F(x)) = \frac{d^2}{dx^2}(\log(1 - F(x)))$. Note that $l_F$ is a measure of "flatness" of function $\log F$. Because of log-concavity of $F$ and $1-F$ (cf. Assumption 3.1), we have $l_F > 0$.

Recall that in the CORP policy, at the beginning of each episode $k > 1$, the preference vectors $\beta_i$ are estimated via optimizing the log-likelihood function corresponding to the outcomes of auctions in the previous episode; see Equation (8). Now, consider buyer $i$ that bids untruthfully in period $t \in E_{k-1}$. Assume for the moment that bids of other buyers in this period is fixed. Then, the untruthful bid of buyer $i$ in period $t \in E_{k-1}$ may influence the estimation of his preference vector in episode $k$ only when his untruthful bid changes the allocation of the item in this period, i.e., $\mathbb{I}(v_{it} > \max\{b_{-it}^+, r_{it}\}) \neq \mathbb{I}(b_{it} > \max\{b_{-it}^+, r_{it}\})$. This is the case because the preference vectors are estimated using the outcome of the auctions and *not* the submitted bids. When $\mathbb{I}(v_{it} > \max\{b_{-it}^+, r_{it}\}) \neq \mathbb{I}(b_{it} > \max\{b_{-it}^+, r_{it}\})$ holds, we say buyer $i$ "lies" in period $t$. For each buyer $i \in [N]$, we further define the set of "lies" in episode $k - 1$, indicated by $\mathsf{L}_{ik}$, as follows:

$$\mathsf{L}_{ik} = \left\{ t : t \in E_{k-1}, \mathbb{I}(v_{it} > \max\{b_{-it}^+, r_{it}\}) \neq \mathbb{I}(b_{it} > \max\{b_{-it}^+, r_{it}\}) \right\}. \tag{17}$$

In other words, $\mathsf{L}_{ik}$ consists of all the periods in episode $k - 1$ in which buyer $i$ lies. We note that the set of lies in episode $k - 1$, $\mathsf{L}_{ik}$, depends on the reserve prices $r_{it}$, $t \in E_{k-1}$, where the reserve prices are (mostly) set using the outcome of the auctions in episode $k - 2$. Because of this dependency, $\mathsf{L}_{ik}$ may also depend on all the submitted bids in episodes $1, 2, \ldots, k - 1$. However, we will show that regardless of the values of $r_{it}$'s, the size of $\mathsf{L}_{ik}$ is logarithmic in the length of episode $k - 1$; see Proposition C.2.

Next, we quantify the adverse effect of lies on the firm's estimates of the preference vectors. In particular, the next proposition provides an upper bound on the estimation error of $\widehat{\beta}_{ik}$ in terms of the number of samples used in the log-likelihood function ($\ell_{k-1}$), the dimension of the feature vector ($d$), and the number of lies ($|L_{ik}|$). Proof of Proposition C.1 is deferred to Section F.

**Proposition C.1** (Impact of Lies on Estimated Preference Vectors)**.** *Let $\widehat{\beta}_{ik}$ be the solution of the optimization problem (8). Then, under Assumption 3.1, there exist constants $c_0$, $c_1$, and $c_2$ such that for $\ell_{k-1} \geq c_0 d$, with probability at least $1 - d^{-0.5}\ell_{k-1}^{-1.5} - 2e^{-c_2\ell_{k-1}}$, we have*

$$\|\widehat{\beta}_{ik} - \beta_i\|^2 \leq \frac{c_1 d^2}{l_F^2} \left( \left( \frac{|\mathsf{L}_{ik}|}{\ell_{k-1}} \right)^2 + \frac{\log(\ell_{k-1}d)}{\ell_{k-1}} \right) \quad i \in [N], \tag{18}$$

*where $l_F$ is defined in Equation (16).*

We note that the estimation error of $\widehat{\beta}_{ik}$'s affects the firm's regret, as reserve prices are set based on these estimates. By Proposition C.1, to keep our estimation errors small, the buyers should not have the incentive to lie in too many periods. In the next proposition, we show that for each episode $k$, the number of lies from a buyer is at most logarithmic in the length of the episode.

There is another way that bidding untruthfully can impact the firm's regret. Recall that in each period $t$, the firm collects the revenue of $\max\{b_t^-, r_t^+\}$ if the highest buyer clears his reserve. Then, by bidding untruthfully, the second highest bid $b_t^-$ may go down. Further, the winner can change, and this, in turn, can lower reserve price of the winner, $r_t^+$. To bound this impact of untruthful bidding, in the following proposition we bound the amount of underbidding from buyers who do not win an auction and the amount of overbidding from buyers who win an auction. Precisely, we show that the total amount of underbidding from each buyer $i$, in all periods $t \in E_k$ that he does not win the auction, is at most logarithmic in the length of that episode. We further show that the total amount of overbidding from each buyer $i$, in all periods $t \in E_k$ that he wins the auction, is at most logarithmic in the length of that episode.

**Proposition C.2** (Bounding the Number of Lies). *Denote by $s_{it}$ and $o_{it}$ the amount of shading and overbidding from buyer $i \in [N]$ in period $t$, i.e., $s_{it} = (v_{it} - b_{it})_+$, and $o_{it} = (b_{it} - v_{it})_+$, where $(y)_+$ is $y$ when $y \geq 0$ and zero otherwise. Then, there exist constants $c_3$, $c_4$, and $c_5$[8], such that for any fixed $0 \leq \delta \leq 1$, with probability at least $1 - \delta/\ell_{k-1}$, the following holds:*

$$|\mathsf{L}_{ik}| \ \leq \ c_3 \log(\ell_{k-1}/\delta) \quad i \in [N]. \tag{19}$$

*Further, we have*

$$\sum_{t \in E_{k-1}} s_{it}(1 - q_{it}) \ \leq \ c_4 \log(\ell_{k-1}) \quad i \in [N], \tag{20}$$

$$\sum_{t \in E_{k-1}} o_{it} q_{it} \ \leq \ c_5 \log(\ell_{k-1}) \quad i \in [N]. \tag{21}$$

Proof of Proposition C.2 is given in Section G. The main idea of the proof is to compute the excess utility that a strategic buyer can earn in the next episodes by bidding untruthfully in the current episode, and compare it with the utility loss that he suffers in the current episode because of his strategic behavior. The result then follows by using the fact that for a utility-maximizing buyer, the net excess utility should be nonnegative.

Up to here, we have established the impact of lies on our estimation, bounded the number of lies and the amount of underbidding from buyers. Next, using these results, we present a lower bound on the expected revenue of our policy in any period $t \in E_k$. We drop the superscript $\pi$ in our notation as it is clear from the context.

For each period $t$, we define a random variable $\xi_t$ that takes values in $\{0, 1\}$, with $\xi_t = 1$ if the firm is in the exploitation phase and $\xi_t = 0$ otherwise. From the description of our policy, for any period $t$ in episode $k$, ($t \in E_k$), we have $\mathbb{P}(\xi_t = 0) = 1/\ell_k$. We first lower bound the firm's expected revenue in an exploitation period $t \in E_k$. Recall that in an exploitation period, the firm runs a second-price auction with reserve. Thus, we have

$$\mathsf{rev}_t \ \geq \ \mathbb{P}(\xi_t = 1)\mathbb{E}[\max\{b_t^-, r_t^+\}\mathbb{I}(b_t^+ \geq r_t^+)], \tag{22}$$

where the expectation is w.r.t. the randomness in the submitted bids. Since in each period $t$, at most one of the buyers gets the item, we can rewrite (22) as follows:

$$\mathsf{rev}_t \ \geq \ \mathbb{P}(\xi_t = 1) \sum_{i=1}^{N} \mathbb{E}\left[\max\{b_t^-, r_{it}\}\mathbb{I}(b_{it} > \max\{b_t^-, r_{it}\})\right]$$

$$= \ \left(1 - \frac{1}{\ell_k}\right) \sum_{i=1}^{N} \mathbb{E}\left[\max\{b_t^-, r_{it}\}\mathbb{I}(b_{it} > \max\{b_t^-, r_{it}\})\right].$$

Next, we compare $\mathsf{rev}_t$ with the expected revenue of the benchmark in period $t$, $\mathsf{rev}_t^\star$. Recalling (5), we have

$$\mathsf{rev}_t^\star \;=\; \sum_{i=1}^{N} \mathbb{E}\left[\max\{v_t^-, r_{it}^\star\} \mathbb{I}(v_{it} > \max\{v_t^-, r_{it}^\star\})\right] . \tag{23}$$

Therefore, the regret of the policy in period $t$ can be upper bounded as

$$\mathsf{rev}_t^\star - \mathsf{rev}_t \;\le\; \left(\frac{1}{\ell_k}\right)\mathsf{rev}_t^\star$$

$$+ \left(1 - \frac{1}{\ell_k}\right)\sum_{i=1}^{N} \mathbb{E}\left[\max\{v_t^-, r_{it}^\star\}\mathbb{I}(v_{it} > \max\{v_t^-, r_{it}^\star\}) - \max\{b_t^-, r_{it}\}\mathbb{I}(b_{it} > \max\{b_t^-, r_{it}\})\right]$$

$$\le \left(\frac{B}{\ell_k}\right) + \left(1 - \frac{1}{\ell_k}\right)\sum_{i=1}^{N} \mathbb{E}\left[\max\{v_t^-, r_{it}^\star\}\mathbb{I}(v_{it} > \max\{v_t^-, r_{it}^\star\}) - \max\{b_t^-, r_{it}\}\mathbb{I}(b_{it} > \max\{b_t^-, r_{it}\})\right],$$
$$\tag{24}$$

where in the last equation, we used the fact that $\mathsf{rev}_t^\star \le B$. We break down the second expression in (24) into two terms:

$$\Delta_{1,t} \;=\; \sum_{i=1}^{N}\left[\max\{v_t^-, r_{it}^\star\}\mathbb{I}(v_{it} > \max\{v_t^-, r_{it}^\star\}) - \max\{v_t^-, r_{it}\}\mathbb{I}(v_{it} > \max\{v_t^-, r_{it}\})\right] \tag{25}$$

$$\Delta_{2,t} \;=\; \sum_{i=1}^{N}\left[\max\{v_t^-, r_{it}\}\mathbb{I}(v_{it} > \max\{v_t^-, r_{it}\}) - \max\{b_t^-, r_{it}\}\mathbb{I}(b_{it} > \max\{b_t^-, r_{it}\})\right] \tag{26}$$

Using our notation, Equation (24) can be rewritten as:

$$\mathsf{rev}_t^\star - \mathsf{rev}_t \le \frac{B}{\ell_k} + \left(1 - \frac{1}{\ell_k}\right)\mathbb{E}[\Delta_{1,t} + \Delta_{2,t}] . \tag{27}$$

In the sequel, we will bound each term $\Delta_{1,t}$ and $\Delta_{2,t}$ separately. But before proceeding, let us pause to explain these terms and the intuition behind their definition. The regret of the firm's policy is due to two factors:

1. **Mismatch between $\beta_i$ and $\widehat{\beta}_{ik}$:** The mismatch between the true preference vectors $\beta_i$ and the estimation $\widehat{\beta}_{ik}$ leads to a difference between the benchmark reserves ($r_{it}^\star$) and the posted reserves by the firm ($r_{it}$). The term $\Delta_{1,t}$ captures this factor and its effect on the regret. We will use Proposition C.1 along with our first result in Proposition C.2 to bound $\Delta_{1,t}$.

2. **Mismatch between $v_t^-$ and $b_t^-$ and change of the winner:** Note that the benchmark revenue $\mathsf{rev}_t^\star$ is measured against truthful buyers, while the firm's revenue under our policy is measured against strategic buyers. The strategic behavior of buyers not only affects the quality of estimates $\widehat{\beta}_{ik}$ (and therefore the reserves $r_{it}$) but it may also affect the firm's revenue via another quite subtle factor. Indeed, due to the strategic behavior of buyers, the second highest bid might go down or the winner of the auction might change from the case of truthful buyers and this may decrease the reserve of the winner. The decrease in the second highest bid or the reserve price of the winner can hurt the firm's revenue. The term $\Delta_{2,t}$ captures these effects. We will use our second result in Proposition C.2 to bound $\Delta_{2,t}$.

**Bounding $\Delta_{1,t}$:** We now move to bounding $\Delta_{1,t}$. Recall that

$$\mathbb{E}[\Delta_{1,t}] = \sum_{i=1}^{N} \mathbb{E}\left[\max\{v_t^-, r_{it}^\star\}\mathbb{I}(v_{it} > \max\{v_t^-, r_{it}^\star\}) - \max\{v_t^-, r_{it}\}\mathbb{I}(v_{it} > \max\{v_t^-, r_{it}\})\right] .$$

Here, the expectation is w.r.t. the randomness in the buyers' valuations. Note that the first expression inside the summation denotes the firm's revenue when buyer $i$ wins the auction with reserve $r_{it}^\star$, while the second expression is the analogous term when the buyer $i$'s reserve is $r_{it}$. Further, conditional on

the feature vector $x_t$, reserves $r_{it}^\star$ and $r_{it}$ are independent of $v_t^-$, and the right-hand side of the last equation can be written in terms of function $W_{it}(r)$, defined below:

$$W_{it}(r) \equiv \mathbb{E}\Big[\max\{v_t^-, r\}\mathbb{I}(v_{it} \geq \max\{v_t^-, r\})\Big|x_t\Big], \tag{28}$$

where the expectation is with respect to valuation noises, conditional on $x_t$. By the law of iterated expectation, we can write $\mathbb{E}[\Delta_{1,t}]$ in terms of $W_{it}(r)$. More specifically, we first take the expectation conditional on $x_t$ and then take the expectation w.r.t. $x_t$.

Hence,

$$
\begin{aligned}
\mathbb{E}[\Delta_{1,t}] &= \mathbb{E}[\mathbb{E}[\Delta_{1,t}|x_t]] \\
&= \sum_{i=1}^{N} \mathbb{E}[W_{it}(r_{it}^\star) - W_{it}(r_{it})] \\
&= \sum_{i=1}^{N} \mathbb{E}\left[W_{it}'(r_{it}^\star)(r_{it}^\star - r_{it}) - \frac{1}{2}W_{it}''(r)(r_{it}^\star - r_{it})^2\right],
\end{aligned}
\tag{29}
$$

for some $r$ between $r_{it}$ and $r_{it}^\star$.[9] We will make use of the following two lemmas to bound the above equation. The proof of all technical lemmas in this section are deferred to Section J.

**Lemma C.3** (Property of Function $W_{it}$). *For the benchmark reserve $r_{it}^\star$, given by (4), and function $W_{it}(r)$, given by (28), we have $W_{it}'(r_{it}^\star) = 0$. Further, for any $r$ between $r_{it}$ and $r_{it}^\star$, we have $|W_{it}''(r)| \leq c$, for a constant $c > 0$.*

**Lemma C.4** (Errors in Reserve Prices). *Let $t \in E_k$ with $\xi_t = 1$. Then, conditioned on the feature vector $x_t$ and $\widehat{\beta}_{ik}$, the following holds:*

$$|r_{it}^\star - r_{it}| \leq |\langle x_t, \beta_i - \widehat{\beta}_{ik}\rangle|, \tag{30}$$

*where $r_{it}^\star$ and $r_{it}$ are defined in (4) and (11), respectively.*

Applying Lemma C.3 in Equation (29), we get

$$
\begin{aligned}
\mathbb{E}[\Delta_{1,t}] &\leq \frac{c}{2}\sum_{i=1}^{N}\mathbb{E}[(r_{it}^\star - r_{it})^2] \\
&\leq \frac{c}{2}\sum_{i=1}^{N}\mathbb{E}\left[\mathbb{E}\Big[(r_{it}^\star - r_{it})^2\Big|x_t, \widehat{\beta}_{ik}\Big]\right] \\
&\leq \frac{c}{2}\sum_{i=1}^{N}\mathbb{E}[\langle x_t, \beta_i - \widehat{\beta}_{ik}\rangle^2],
\end{aligned}
\tag{31}
$$

where in the last step, we employed Lemma C.4. We next further simplify the r.h.s. of the last equation. By using the fact that our estimate $\widehat{\beta}_{ik}$ is constructed using samples from the previous episode and consequently is independent from the current feature $x_t$, we get

$$\mathbb{E}[\langle x_t, \beta_i - \widehat{\beta}_{ik}\rangle^2] = \mathbb{E}[\langle \beta_i - \widehat{\beta}_{ik}, \Sigma(\beta_i - \widehat{\beta}_{ik})\rangle] \leq \frac{c_{\max}}{d}\mathbb{E}[\|\beta_i - \widehat{\beta}_{ik}\|^2], \tag{32}$$

where $\Sigma = \mathbb{E}[x_t x_t^\mathsf{T}]$ is the second-moment matrix of features $x_t$, and $c_{\max}/d$ is the bound on the maximum eigenvalue of covariance $\Sigma$. [10] Here, the first inequality follows from taking the expectation w.r.t. $x_t$ and using the fact that $x_t$ and $\widehat{\beta}_{ik}$ are independent; the second inequality follows from the definition of the maximum eigenvalue.

Putting Equations (31) and (32) together, we get

$$\mathbb{E}[\Delta_{1,t}] \leq \frac{c'}{d}\sum_{i=1}^{N}\mathbb{E}[\|\beta_i - \widehat{\beta}_{ik}\|^2]. \tag{33}$$

Here, $c' = \frac{1}{2}cc_{\max}$.

**Bounding $\Delta_{2,t}$:** We next proceed with bounding $\Delta_{2,t}$. To do so, we use the following preliminary lemma.

**Lemma C.5.** *Let $v_t^-$ and $b_t^-$, respectively, denote the second highest valuation and the second highest bid submitted by the buyers in the CORP policy. Denote by $s_{it}$ and $o_{it}$ the amount of shading and overbidding from buyer $i \in [N]$ in period $t$, i.e., $s_{it} = (v_{it} - b_{it})_+$, and $o_{it} = (b_{it} - v_{it})_+$. Then,*

$$(v_t^- - b_t^-)_+ \leq \max\left\{s_{it}(1 - q_{it}) : i \in [N]\right\}. \tag{34}$$

*Further, for any buyer $i$ with $q_{it} = 0$, the following holds:*

$$(b_{-it}^+ - v_{-it}^+)_+ \leq \max\left\{o_{jt}q_{jt} : j \in [N], j \neq i\right\}. \tag{35}$$

Proof of Lemma C.5 is given in Section J.3.

Note that $\Delta_{2,t}$, given by (26), can be written as

$$\Delta_{2,t} = \sum_{i=1}^{N}\left[\max\{v_t^-, r_{it}\}\mathbb{I}(v_{it} > \max\{v_{-it}^+, r_{it}\}) - \max\{b_t^-, r_{it}\}\mathbb{I}(b_{it} > \max\{b_{-it}^+, r_{it}\})\right]. \tag{36}$$

Define $\mathsf{L}_{k+1} = \cup_{i=1}^{N}\mathsf{L}_{i(k+1)}$, where $\mathsf{L}_{i(k+1)}$, given by (17), denotes the set of periods in episode $k$ that buyer $i$ lies. For $t \in \mathsf{L}_{k+1}$, we avail the trivial bound

$$\Delta_{2,t} \leq B, \tag{37}$$

which is true because the revenue of the benchmark in any period $t$ is at most $v_t^+ \leq B$. For $t \notin \mathsf{L}_{k+1}$, we have $\mathbb{I}(b_{it} > \max\{b_{-it}^+, r_{it}\}) = \mathbb{I}(v_{it} > \max\{b_{-it}^+, r_{it}\})$, for all $i \in [N]$. Therefore, we can write

$$\mathbb{E}[\Delta_{2,t}\,\mathbb{I}(t \notin \mathsf{L}_{k+1})] =$$
$$\sum_{i=1}^{N}\mathbb{E}\left[\max\{v_t^-, r_{it}\}\mathbb{I}(v_{it} > \max\{v_{-it}^+, r_{it}\}) - \max\{b_t^-, r_{it}\}\mathbb{I}(v_{it} > \max\{b_{-it}^+, r_{it}\})\right]. \tag{38}$$

To bound the r.h.s of (38), we use the fact that for any two indicators $\chi_1, \chi_2$ and any $a, b \geq 0$, we have $a\chi_1 - b\chi_2 \leq (a - b)\chi_2 + a\chi_1(1 - \chi_2)$. Applying this inequality to (38) with $\chi_1 = \mathbb{I}(v_{it} > \max\{v_{-it}^+, r_{it}\})$, $\chi_2 = \mathbb{I}(v_{it} > \max\{b_{-it}^+, r_{it}\})$, $a = \max\{v_t^-, r_{it}\}$, and $b = \max\{b_t^-, r_{it}\}$, we get

$$\mathbb{E}[\Delta_{2,t}\,\mathbb{I}(t \notin \mathsf{L}_{k+1})]$$
$$\leq \sum_{i=1}^{N}\mathbb{E}\left[(\max\{v_t^-, r_{it}\} - \max\{b_t^-, r_{it}\})\mathbb{I}(v_{it} > \max\{b_{-it}^+, r_{it}\})\right]$$
$$+ \sum_{i=1}^{N}\mathbb{E}\left[\max\{v_t^-, r_{it}\}\mathbb{I}\left(\max\{v_{-it}^+, r_{it}\} < v_{it} < \max\{b_{-it}^+, r_{it}\}\right)\right].$$

Then, by using the fact that $\max\{a, c\} - \max\{b, c\} \leq (a - b)_+$, we get
$$\mathbb{E}[\Delta_{2,t}\,\mathbb{I}(t \notin \mathsf{L}_{k+1})]$$

$$\leq \sum_{i=1}^{N}\mathbb{E}\left[(v_t^- - b_t^-)_+\mathbb{I}(v_{it} > \max\{b_{-it}^+, r_{it}\})\right] + \sum_{i=1}^{N}\mathbb{E}\left[\max\{v_t^-, r_{it}\}\mathbb{I}\left(\max\{v_{-it}^+, r_{it}\} < v_{it} < \max\{b_{-it}^+, r_{it}\}\right)\right]$$

$$\leq \mathbb{E}\left[(v_t^- - b_t^-)_+ \sum_{i=1}^{N}\mathbb{I}(v_{it} > \max\{b_{-it}^+, r_{it}\})\right] + B\sum_{i=1}^{N}\mathbb{P}\left(\max\{v_{-it}^+, r_{it}\} < v_{it} < \max\{b_{-it}^+, r_{it}\}\right)$$

$$= \mathbb{E}\left[(v_t^- - b_t^-)_+ \sum_{i=1}^{N} q_{it}\right] + B\sum_{i=1}^{N}\mathbb{P}\left(\max\{v_{-it}^+, r_{it}\} < v_{it} < \max\{b_{-it}^+, r_{it}\}\right)$$

$$\leq \mathbb{E}[(v_t^- - b_t^-)_+] + B\sum_{i=1}^{N}\mathbb{P}\left(\max\{v_{-it}^+, r_{it}\} < v_{it} < \max\{b_{-it}^+, r_{it}\}\right).$$

Here, in the second inequality we used $\max\{v_t^-, r_{it}\} \leq B$. In the equality thereafter, we used the fact that $t \notin \mathsf{L}_{k+1}$ and hence $\mathbb{I}(v_{it} > \max\{b_{-it}^+, r_{it}\}) = \mathbb{I}(b_{it} > \max\{b_{-it}^+, r_{it}\}) \equiv q_{it}$. The last inequality holds since the item can be allocated to at most one buyer and hence $\sum_{i=1}^N q_{it} \leq 1$. We next bound the first term by virtue of Lemma C.5 (Equation (34)). Specifically,

$$(v_t^- - b_t^-)_+ \;\leq\; \max\{s_{it}(1 - q_{it}) : i \in [N]\} \;\leq\; \sum_{i=1}^N s_{it}(1 - q_{it}). \tag{39}$$

To bound the second term, we again use inequality that $\max\{a, c\} - \max\{b, c\} \leq (a - b)_+$ with $a = b_{-it}^+$, $b = v_{-it}^+$, and $c = r_{it}$:

$$\mathbb{P}\big(\max\{v_{-it}^+, r_{it}\} < v_{it} < \max\{b_{-it}^+, r_{it}\} \big| b_{-it}^+, v_{-it}^+\big)$$
$$\leq \mathbb{P}\big(\max\{b_{-it}^+, r_{it}\} - (\underline{\phantom{v}}v_{-it}^+)_+ < v_{it} < \max\{b_{-it}^+, r_{it}\} \big| b_{-it}^+, v_{-it}^+\big)$$
$$= \mathbb{P}\big(\max\{b_{-it}^+, r_{it}\} - (\underline{\phantom{v}}v_{-it}^+)_+ - \langle x_t, \beta_i\rangle < z_{it} < \max\{b_{-it}^+, r_{it}\} - \langle x_t, \beta_i\rangle \big| b_{-it}^+, v_{-it}^+\big)$$
$$= \int_{\max\{b_{-it}^+, r_{it}\} - (\underline{\phantom{v}}v_{-it}^+)_+ - \langle x_t, \beta_i\rangle}^{\max\{b_{-it}^+, r_{it}\} - \langle x_t, \beta_i\rangle} f(z)\mathrm{d}z \;<\; \hat{c}(\underline{\phantom{v}}v_{-it}^+)_+. \tag{40}$$

The first equality follows readily by substituting for $v_{it} = \langle x_t, \beta_i\rangle + z_{it}$. In addition, in the last equality, $\hat{c} \equiv \max_{v \in [-B_n, B_n]} f(v)$ is the bound on the noise density,[11] and this equality holds because $z_{it}$ is independent of $v_{-it}^+$, $b_{-it}^+$, reserve $r_{it}$, and the feature vector $x_t$. We point out that when $\mathbb{I}(\max\{v_{-it}^+, r_{it}\} < v_{it} < \max\{b_{-it}^+, r_{it}\}) = 1$, buyer $i$ does not win the item. To see this, recall that we compute the probability of $\mathbb{I}(\max\{v_{-it}^+, r_{it}\} < v_{it} < \max\{b_{-it}^+, r_{it}\})$ when $t \notin \mathsf{L}_{k+1}$. This implies that $\mathbb{I}(b_{it} > \max\{b_{-it}^+, r_{it}\}) = \mathbb{I}(v_{it} > \max\{b_{-it}^+, r_{it}\})$ and as a result when $\mathbb{I}(\max\{v_{-it}^+, r_{it}\} < v_{it} < \max\{b_{-it}^+, r_{it}\}) = 1$, buyer $i$ does not win, i.e., $q_{it} = 0$. The fact $q_{it} = 0$ enables us to use Lemma C.5 (Equation (35)) along with Equation (40) to get

$$\mathbb{P}\big(\max\{v_{-it}^+, r_{it}\} < v_{it} < \max\{b_{-it}^+, r_{it}\} \big| b_{-it}^+, v_{-it}^+\big)$$

$$\leq \hat{c}\max\{o_{jt}q_{jt} : j \in [N], j \neq i\} \leq \hat{c}\sum_{j=1}^N o_{jt}q_{jt}. \tag{41}$$

Putting bounds in Equations (37), (39), (41) together, we have

$$\mathbb{E}[\Delta_{2,t}] = \mathbb{E}[\Delta_{2,t}\mathbb{I}(t \in \mathsf{L}_{k+1})] + \mathbb{E}[\Delta_{2,t}\mathbb{I}(t \notin \mathsf{L}_{k+1})]$$

$$\leq B\mathbb{P}(t \in \mathsf{L}_{k+1}) + \mathbb{E}\Big[\sum_{i=1}^N s_{it}(1 - q_{it}) + \hat{c}B\sum_{j=1}^N o_{jt}q_{jt}\Big]. \tag{42}$$

**Combining bounds on $\Delta_{1,t}$ and $\Delta_{2,t}$:** To summarize, using bounds (33) and (42) in Equation (27), for all $t \in E_k$, we have

$$\mathsf{rev}_t^\star - \mathsf{rev}_t \;\leq\; \frac{B}{\ell_k} + \Big(1 - \frac{1}{\ell_k}\Big)\mathbb{E}[\Delta_{1,t} + \Delta_{2,t}]$$

$$\leq \frac{B}{\ell_k} + \frac{c'}{d}\sum_{i=1}^N \mathbb{E}[\|\beta_i - \widehat{\beta}_{ik}\|^2] + B\mathbb{P}(t \in \mathsf{L}_{k+1}) + \mathbb{E}\Big[\sum_{i=1}^N s_{it}(1 - q_{it}) + \hat{c}B\sum_{i=1}^N o_{it}q_{it}\Big].$$
$$\tag{43}$$

We are now ready to bound the total regret of our policy. Since the length of episodes doubles each time, the number of episodes up to time $t$ would be at most $K = \lfloor \log T \rfloor + 1$. We then have

$$\mathsf{Reg}(T) \;\leq\; \sum_{k=1}^K \mathsf{Reg}_k, \tag{44}$$

where $\mathsf{Reg}_k$ is the regret of our policy in episode $k \in [K]$. We bound the total regret over each episode by considering the following two cases: $\ell_{k-1} \leq c_0 d$ and $\ell_{k-1} > c_0 d$. Here, $c_0$ is the constant in the statement of Proposition C.1.

- **Case 1:** $\ell_{k-1} \leq c_0 d$: In this case, we use the trivial bound $\mathsf{rev}_t^\star - \mathsf{rev}_t \leq \mathsf{rev}_t^\star \leq v_t^+ \leq B$. Given that the length of episode $k$ is $\ell_k \leq 2c_0 d$, the total lengths of all such episodes is at most $4c_0 d$ and therefore, the total regret over such episodes is at most $4c_0 Bd$.

- **Case 2:** $\ell_{k-1} > c_0 d$: In that case, we use bound (43) on the regret in each period of episode $k$:

$$
\begin{aligned}
\mathsf{Reg}_k \;=\;& \sum_{t \in E_k} (\mathsf{rev}_t^\star - \mathsf{rev}_t) \\
\leq\;& \frac{B}{\ell_k}\ell_k + \frac{c'}{d}\ell_k \sum_{i=1}^{N} \mathbb{E}[\|\beta_i - \widehat{\beta}_{ik}\|^2] + B\mathbb{E}[|\mathsf{L}_{k+1}|] + \sum_{i=1}^{N} \mathbb{E}\Big[ \sum_{t \in E_k} s_{it}(1-q_{it}) + \hat{c}B \sum_{t \in E_k} o_{it}q_{it} \Big].
\end{aligned}
$$
(45)

We treat each term on the right-hand side of (45) separately.

We first bound the second term, i.e., $c'\ell_k \sum_{i=1}^{N} \mathbb{E}[\|\beta_i - \widehat{\beta}_{ik}\|^2]$. Define the probability event $\mathcal{G}$, such that event $\mathcal{G}$ happens when Equations (18) and (19) hold; that is, the number of lies satisfies (19) and the estimation errors satisfies (18). By Proposition C.1 and C.2, the probability of complement of event $\mathcal{G}$, denoted by $\mathcal{G}^c$, is given by

$$
\mathbb{P}(\mathcal{G}^c) \leq \frac{\delta}{\ell_{k-1}} + d^{-0.5}\ell_{k-1}^{-1.5} + 2e^{-c_2 \ell_{k-1}} \,.
$$

Using these propositions again, we get

$$
\begin{aligned}
\mathbb{E}[\|\beta_i - \widehat{\beta}_{ik}\|^2] \;=\;& \mathbb{E}[\|\beta_i - \widehat{\beta}_{ik}\|^2 \,\mathbb{I}(\mathcal{G})] + \mathbb{E}[\|\beta_i - \widehat{\beta}_{ik}\|^2 \,\mathbb{I}(\mathcal{G}^c)] \\
\leq\;& \frac{c_1 d^2}{l_F{}^2}\left( \left(\frac{c_3 \log(\ell_{k-1}/\delta)}{\ell_{k-1}}\right)^2 + \frac{\log(\ell_{k-1}d)}{\ell_{k-1}} \right) + 4B^2 \mathbb{P}(\mathcal{G}^c) \\
\leq\;& c_6 \left( \left(\frac{d\log(T/\delta)}{\ell_{k-1}}\right)^2 + \frac{d^2 \log(Td)}{\ell_{k-1}} + \frac{\delta}{\ell_{k-1}} + d^{-0.5}\ell_{k-1}^{-1.5} + e^{-c_2 \ell_{k-1}} \right),
\end{aligned}
$$
(46)

where we absorb various constants into constant $c_6$ and used $\ell_{k-1} \leq T$.

Regarding the third term, i.e., $B\mathbb{E}[|\mathsf{L}_{k+1}|]$, by Proposition C.2 we have

$$
\mathbb{E}[|\mathsf{L}_{k+1}|] \;\leq\; \sum_{i=1}^{N} \mathbb{E}[|\mathsf{L}_{i,k+1}|] \;\leq\; Nc_3 \log(\ell_k/\delta)\left(1 - \frac{\delta}{\ell_k}\right) + N\ell_k \frac{\delta}{\ell_k} \;\leq\; Nc_3 \log(\ell_k/\delta) + N\delta,
$$
(47)

where $\delta$ and $c_3$ are defined in Proposition C.2.

Finally, we bound the last term of Equation (45). Invoking Equations (20) and (21), we have

$$
\sum_{i=1}^{N} \mathbb{E}\Big[ \sum_{t \in E_k} s_{it}(1-q_{it}) + \hat{c}B \sum_{t \in E_k} o_{it}q_{it} \Big] \leq (c_4 + \hat{c}c_5 B)N \log(\ell_k).
$$
(48)

We employ bounds (47), (46) and (48) in bound (45) and keep only the dominant terms, from which we get

$$
\mathsf{Reg}_k \leq c_7 d \left( \log^2(T)\frac{1}{\ell_k} + \log(Td) \right),
$$
(49)

for a constant $c_7$ that depends on $N,\ B,\ M,\ \delta$, and $\gamma$.

As the final step, we combine our regent bounds for the two cases to find the total regret of our policy. Let $K_1 = \lfloor \log(c_0 d) \rfloor + 3$ be an upper bound on the number of episodes that fall into Case 1. Also, recall that $K = \lfloor \log T \rfloor + 1$ is the upper bound on the number of episodes up to time $t$. Then, by

Equation (49), we obtain

$$
\begin{aligned}
\mathsf{Reg}(T) \;\leq\; & 4c_0 Bd + c_7 d \left( \log^2(T) \sum_{k=K_1}^{K} \frac{1}{\ell_k} + K \log(Td) \right) \\
\leq\; & 4c_0 Bd + c_7 d \left( \log^2(T) \sum_{k=K_1}^{K} \frac{1}{2^{k-1}} + K \log(Td) \right) \\
\leq\; & 4c_0 Bd + c_7 d \left( \log^2(T) + \log(T) \log(Td) \right) ,
\end{aligned}
$$

which completes the proof.

## D  Proof of Theorem B.3

The proof, in sprit, is similar to that of Theorem 4.1. We first state an upper bound on the estimation error of the preference vectors $\beta_i$. This proposition is analogous to Proposition C.1, where instead of log-likelihood estimator, we use the least square estimator.

**Proposition D.1** (Impact of Lies on Estimated Preference Vectors in SCORP). *Suppose that Assumption B.1 holds and let $\widehat{\beta}_{ik}$ be the solution of optimization* (13). *Then, there exist constants $c_0$, $c_1$, and $c_2$ such that for $\ell_{k-1} \geq c_0 d$, with probability at least $1 - d^{-0.5} \ell_{k-1}^{-1.5} - 2e^{-c_2 |I_k|}$, we have*

$$
\| \widehat{\beta}_{ik} - \beta_i \|^2 \leq c_1 d^2 \left( \left( \frac{|\mathsf{L}_{ik}|}{|I_k|} \right)^2 + \frac{\log(\ell_{k-1} d)}{|I_k|} \right) \quad i \in [N] , \tag{50}
$$

*where $\mathsf{L}_{ik}$ is the set of lies associated to buyer $i$ in episode $k$, given by* (17)*, and $I_k$ is the set of pure exploration periods in episode $k$.*

Proof of Proposition D.1 is given in Section H. We next proceed to bound the number of lies $|\mathsf{L}_{ik}|$. We argue that the same bound given in Proposition C.2 still holds for SCORP.

**Proposition D.2** (Bounding the Number of Lies in SCORP). *Denote by $s_{it}$ and $o_{it}$ the amount of shading and overbidding from buyer $i \in [N]$ in period $t$, i.e., $s_{it} = (v_{it} - b_{it})_+$, and $o_{it} = (b_{it} - v_{it})_+$. Then, there exists a constant $c_3$ such that for any fixed $0 \leq \delta \leq 1$, with probability at least $1 - \delta/\ell_{k-1}$, the following holds:*

$$
|\mathsf{L}_{ik}| \;\leq\; c_3 \log(\ell_{k-1}/\delta) \quad i \in [N] . \tag{51}
$$

Similar to Proposition C.2, we prove Proposition D.2 by balancing the utility loss of an untruthful buyer with his future utility gain. The proof is presented in Section I.

Our next lemma relates the difference between the reserves $r_{it}$, set by SCORP policy, and benchmark reserves $r_{it}^\star$, to the estimation error of preference vectors. This lemma is analogous to Lemma C.4.

**Lemma D.3** (Errors in Reserve Prices). *For $r_{it}^\star$ and $r_{it}$ given by* (12) *and* (15)*, respectively, conditioned on the feature vector $x_t$ and $\widehat{\beta}_{ik}$, the following holds*

$$
|r_{it}^\star - r_{it}| \leq |\langle x_t, \beta_i - \widehat{\beta}_{ik} \rangle| . \tag{52}
$$

We refer to Section J for the proof of Lemma D.3.

Having established the preliminary results, we proceed to bound the regret of SCORP. The proof goes along the same lines of the proof of Theorem 4.1. We fix $k \geq 1$ and focus on the total regret during episode $k$. For the pure exploration phase, i.e., $t \in I_k$, we use the trivial bound $\mathsf{rev}_t^\star - \mathsf{rev}_t \leq B$, which holds since $\mathsf{rev}_t^\star \leq v_t^+ \leq B$.

To bound the regret in periods of the exploitation phase ($t \in E_k \backslash I_k$), we note that SCORP does not use any of the submitted bids during the exploitation phases to estimate the preference vectors, and buyers are cognizant of this point as the seller's learning policy is fully known to them. In addition, since the second-price auctions are strategy-proof, this means that in the exploitation phase, there is

no incentive for buyers to be untruthful. [12]. Hence, for $t \in E_k \backslash I_k$, we have

$$\text{rev}_t = \sum_{i=1}^{N} \mathbb{E}\left[\max\{v_t^-, r_{it}\}\mathbb{I}(v_{it} > \max\{v_t^-, r_{it}\})\right]. \tag{53}$$

This leads to

$$\text{rev}_t^\star - \text{rev}_t = \mathbb{E}\left(\sum_{i=1}^{N}\left[\max\{v_t^-, r_{it}^\star\}\mathbb{I}(v_{it} > \max\{v_t^-, r_{it}^\star\}) - \max\{v_t^-, r_{it}\}\mathbb{I}(v_{it} > \max\{v_t^-, r_{it}\})\right]\right),$$

where the expectation is with respect to the true underlying noise distribution, which can vary over time and is of course, unknown to the firm and the benchmark policy. To bound $(\text{rev}_t^\star - \text{rev}_t)$, we first write it in terms of function $W_{it}(r)$, defined by (28). By virtue of the mean-value theorem, we have

$$\text{rev}_t^\star - \text{rev}_t = \sum_{i=1}^{N} \mathbb{E}[W_{it}(r_{it}^\star) - W_{it}(r_{it})] = \sum_{i=1}^{N} \mathbb{E}[W_{it}'(r)(r_{it}^\star - r_{it})], \tag{54}$$

for some $r$ between $r_{it}$ and $r_{it}^\star$. It is worth noting that, in contrast to Equation (29), here we do not go with Taylor's expansion of order two. The reason is that here $W_{it}'(r_{it}^\star) \neq 0$, because $W_{it}(r)$ is defined based on the *true* unknown noise distribution, while $r_{it}^\star$ is the optimal reserve for the *worst-case* distribution in ambiguity set $\mathcal{F}$. Similar to Lemma C.3, it is straightforward to see that $|W_{it}'(r)| \leq \tilde{c}$ for some constant $\tilde{c} > 0$. Therefore, continuing from Equation (54), we have

$$\begin{aligned}
\text{rev}_t^\star - \text{rev}_t &\leq \tilde{c}\sum_{i=1}^{N} \mathbb{E}[|r_{it}^\star - r_{it}|] = \tilde{c}\sum_{i=1}^{N} \mathbb{E}\left[\mathbb{E}\left[|r_{it}^\star - r_{it}| \Big| x_t, \widehat{\beta}_{ik}\right]\right] \\
&\leq \tilde{c}\sum_{i=1}^{N} \mathbb{E}[|\langle x_t, \beta_i - \widehat{\beta}_{ik}\rangle|] \leq \tilde{c}\sum_{i=1}^{N} \mathbb{E}[\langle x_t, \beta_i - \widehat{\beta}_{ik}\rangle^2]^{1/2} \\
&\leq \frac{c'}{\sqrt{d}}\sum_{i=1}^{N} \mathbb{E}[\|\beta_i - \widehat{\beta}_{ik}\|^2]^{1/2},
\end{aligned} \tag{55}$$

with $c' = \tilde{c}\sqrt{c_{\max}}$. Here, the second inequality holds due to Lemma D.3; the third inequality follows from Cauchy-Schwartz inequality, and the last step is derived as in Equation (32).

We are now ready to bound the total regret up to time $T$. Given that the length of episodes double each time, letting $K = \lfloor \log T \rfloor + 1$, we have $\text{Reg}(T) \leq \sum_{k=1}^{K} \text{Reg}_k$. Similar to the proof of Theorem 4.1, we bound the total regret over each episode by considering two cases:

- **Case 1:** $\ell_{k-1} \leq c_0 d$: Here, $c_0$ is the constant in the statement of Proposition D.1. In this case, as we argued in the proof of Theorem 4.1, the total regret over such episodes is at most $4c_0 B d$.

- **Case 2:** $\ell_{k-1} > c_0 d$: Define event $\mathcal{G}$ such that event $\mathcal{G}$ happens when equations (50) and (19) hold. By Proposition D.1 and D.2, we have

$$\mathbb{P}(\mathcal{G}^c) \leq \frac{\delta}{\ell_{k-1}} + d^{-0.5}\ell_{k-1}^{-1.5} + 2e^{-c_2|I_k|}.$$

Therefore,

$$\begin{aligned}
\mathbb{E}[\|\beta_i - \widehat{\beta}_{ik}\|^2] &= \mathbb{E}[\|\beta_i - \widehat{\beta}_{ik}\|^2 \, \mathbb{I}(\mathcal{G})] + \mathbb{E}[\|\beta_i - \widehat{\beta}_{ik}\|^2 \, \mathbb{I}(\mathcal{G}^c)] \\
&\leq c_1 d^2 \left(\frac{\log(\ell_{k-1}d)}{|I_k|} + \left(\frac{c_3 \log(\ell_{k-1}/\delta)}{|I_k|}\right)^2\right) + 4B^2 \mathbb{P}(\mathcal{G}^c) \\
&\leq c_6 \left(\frac{d^2 \log(Td)}{|I_k|} + \left(\frac{d \log(T/\delta)}{|I_k|}\right)^2 + \frac{\delta}{\ell_{k-1}} + d^{-0.5}\ell_{k-1}^{-1.5} + e^{-c_2|I_k|}\right), \tag{56}
\end{aligned}$$

where we absorb various constants into $c_6$ and used $\ell_{k-1} \le T$.

We next employ bound (56) in Equation (55). By keeping the dominant terms and following the same argument of Equation (49), we get

$$\sum_{t \in E_k \setminus I_k} (\mathsf{rev}_t^\star - \mathsf{rev}_t) \le \hat{c}_6 \sqrt{d} \left( \sqrt{\frac{\log(Td)}{|I_k|}} \ell_k + \frac{\log(T)}{|I_k|} \ell_k \right), \tag{57}$$

for a constant $\hat{c}_6$ that depends on $N$, $B$, $\delta$, and $\gamma$.

Adding the total regret during the pure exploration phase, we obtain

$$\mathsf{Reg}_k = \sum_{t \in I_k} (\mathsf{rev}_t^\star - \mathsf{rev}_t) + \sum_{t \in E_k \setminus I_k} (\mathsf{rev}_t^\star - \mathsf{rev}_t)$$

$$\le B|I_k| + \hat{c}_6 \left( \sqrt{\frac{\log(Td)}{|I_k|}} \ell_k + \frac{\log(T)}{|I_k|} \ell_k \right). \tag{58}$$

Finally, we are ready to bound the cumulative regret up to time $T$. Let $K_1 = \lfloor \log(c_0 d) \rfloor + 3$. Reconciling the above two cases into Equation (58), and substituting for $|I_k| = \lceil \ell_k^{2/3} \rceil$, we obtain

$$\mathsf{Reg}(T) \le 4c_0 Bd + \sum_{k=K_1}^{K} B|I_k| + \hat{c}_6 \sqrt{d} \left( \sqrt{\log(Td)} \sum_{k=K_1}^{K} \frac{\ell_k}{\sqrt{|I_k|}} + \log(T) \sum_{k=K_1}^{K} \frac{\ell_k}{|I_k|} \right)$$

$$\le 4c_0 Bd + B \sum_{k=K_1}^{K} 2^{\frac{2(k-1)}{3}} + \hat{c}_6 \sqrt{d} \left( \sqrt{\log(Td)} \sum_{k=K_1}^{K} 2^{\frac{2(k-1)}{3}} + \log(T) \sum_{k=K_1}^{K} 2^{\frac{k-1}{3}} \right)$$

$$\le 4c_0 Bd + BT^{2/3} + \hat{c}_6 \sqrt{d} \left( \sqrt{\log(Td)} \, T^{2/3} + \log(T) \, T^{1/3} \right),$$

which completes the proof.

## E  Proof of Proposition 3.2

We restate the definition of function $W_{it}(r)$, given by Equation (28):

$$W_{it}(r) \equiv \mathbb{E} \left[ \max\{v_t^-, r\} \mathbb{I}(v_{it} \ge \max\{v_t^-, r\}) \Big| x_t \right]$$

$$= \mathbb{E} \left[ \max\{v_{-it}^+, r\} \mathbb{I}(v_{it} \ge \max\{v_{-it}^+, r\}) \Big| x_t \right],$$

where the expectation is with respect to valuation noises, conditional on $x_t$, and the equality holds because $v_t^- = v_{-it}^+$ when $\mathbb{I}(v_{it} \ge \max\{v_{-it}^+, r\}) = 1$. Note that $W_{it}(r)$ is the firm's revenue in period $t$, when buyer $i$ wins the auction with reserve price $r$.

Let $H_{it}$ be the distribution of $v_{-it}^+$ for fixed $x_t$ and denote by $h_{it}$ its density. The specific form of $H_{it}$ does not matter for the sake of our proof. We have

$$W_{it}(r) = \mathbb{E} \left[ \mathbb{I}(v_{it} > v_{-it}^+ > r) v_{-it}^+ + r \mathbb{I}(v_{it} > r > v_{-it}^+) \Big| x_t \right]$$

$$= \mathbb{E} \left[ \mathbb{I}(\langle x_t, \beta_i \rangle + z_{it} > v_{-it}^+ > r) v_{-it}^+ + r \mathbb{I}(\langle x_t, \beta_i \rangle + z_{it} > r > v_{-it}^+) \Big| x_t \right]$$

$$= \int_r^\infty v h_{it}(v)(1 - F(v - \langle x_t, \beta_i \rangle)) \mathrm{d}v + r H_{it}(r)(1 - F(r - \langle x_t, \beta_i \rangle)). \tag{59}$$

By definition, the optimal reserve price of buyer $i$, denoted by $r_{it}^\star$, is the maximizer of $W_{it}(r)$. By setting the derivative with respect to $r$ equal to zero, we get

$$W_{it}'(r) = H_{it}(r) \Big( (1 - F(r - \langle x_t, \beta_i \rangle)) - r f(r - \langle x_t, \beta_i \rangle) \Big) = 0, \tag{60}$$

which implies that the optimal price $r_{it}^\star$ should satisfy

$$1 - F(r - \langle x_t, \beta_i \rangle) = r f(r - \langle x_t, \beta_i \rangle). \tag{61}$$

Now it is easy to see that Equation (61) is also the stationary condition for the function $y(1 - F(y - \langle x_t, \beta_i \rangle))$. Since $1 - F$ is log-concave by Assumption 3.1, function $y \mapsto y(1 - F(y - \langle x_t, \beta_i \rangle))$ is also strictly log-concave for $y > 0$.[13] Therefore, the stationary condition for $y(1 - F(y - \langle x_t, \beta_i \rangle))$ gives its unique global maximum and the proof is complete.[14]

# F    Proof of Proposition C.1

Recall that $\widehat{\beta}_{ik} \in \mathbb{R}^d$ is the solution to the optimization problem (10). By the second-order Taylor's theorem, expanding around $\beta_i$, we have

$$\mathcal{L}_{ik}(\beta_i) - \mathcal{L}_{ik}(\widehat{\beta}_{ik}) = -\langle \nabla \mathcal{L}_{ik}(\beta_i), \widehat{\beta}_{ik} - \beta_i \rangle - \frac{1}{2} \langle \widehat{\beta}_{ik} - \beta_i, \nabla^2 \mathcal{L}_{ik}(\tilde{\beta})(\widehat{\beta}_{ik} - \beta_{ik}) \rangle , \quad (62)$$

for some $\tilde{\beta}$ on the segment connecting $\beta_i$ and $\widehat{\beta}_{ik}$. Throughout, $\nabla \mathcal{L}_{ik}$ and $\nabla^2 \mathcal{L}_{ik}$ respectively denote the gradient and the Hessian of $\mathcal{L}_{ik}$. In the following, we bound $\|\widehat{\beta}_{ik} - \beta_i\|^2$ by bounding the gradient and the Hessian of $\mathcal{L}_{ik}$.

We start with computing the gradient and the Hessian of the loss function $\mathcal{L}_{ik}(\beta)$:

$$\nabla \mathcal{L}_{ik}(\beta) = \frac{1}{\ell_{k-1}} \sum_{t \in E_{k-1}} \mu_{it}(\beta) x_t , \quad \nabla^2 \mathcal{L}_{ik}(\beta) = \frac{1}{\ell_{k-1}} \sum_{t \in E_{k-1}} \eta_{it}(\beta) x_t x_t^{\mathsf{T}} . \quad (63)$$

Here, letting $w_{it}(\beta) = \max\{b^+_{-it}, r_{it}\} - \langle x_t, \beta \rangle$, the term $\mu_{it}(\beta)$ is given by

$$\mu_{it}(\beta) = q_{it} \frac{f(w_{it}(\beta))}{1 - F(w_{it}(\beta))} - (1 - q_{it}) \frac{f(w_{it}(\beta))}{F(w_{it}(\beta))} \quad (64)$$

$$= - q_{it} \log' (1 - F(w_{it}(\beta))) - (1 - q_{it}) \log' (F(w_{it}(\beta))) , \quad (65)$$

where $\log' F(y)$ is the derivative of $\log F(y)$ with respect to $y$.[15] Further, the term $\eta_{it}(\beta)$ is given by

$$\eta_{it}(\beta) = -q_{it} \log'' (1 - F(w_{it}(\beta))) - (1 - q_{it}) \log'' (F(w_{it}(\beta))) . \quad (66)$$

We are now ready to provide an upper bound and a lower bound on the gradient and Hessian of the loss function. These bounds will be used in bounding the estimation error of preference vectors, which is the main goal of this proposition.

**Lemma F.1.** *Define the probability event*

$$\mathcal{E} \equiv \left\{ \|\nabla \mathcal{L}_{ik}(\beta_i)\| \le \lambda_0 \right\}, \quad \text{with } \lambda_0 \equiv 2u_F \sqrt{\frac{\log(\ell_{k-1}d)}{\ell_{k-1}}} + 2u_F \frac{|\mathsf{L}_{ik}|}{\ell_{k-1}} , \quad (67)$$

*where constant $u_F$ is given by*

$$u_F \equiv \sup_{|x| \le B_n} \left\{ \max \left\{ \log' F(x), -\log'(1 - F(x)) \right\} \right\} . \quad (68)$$

*Then, we have $\mathbb{P}(\mathcal{E}) \ge 1 - d^{-0.5} \ell_{k-1}^{-1.5}$. Moreover, we have the following lower bound on the Hessian:*

$$\nabla^2 \mathcal{L}_{ik}(\beta) \succeq l_F \left( \frac{1}{\ell_{k-1}} \sum_{t \in E_{k-1}} x_t x_t^{\mathsf{T}} \right), \quad \text{for all } \|\beta\| \le B , \quad (69)$$

*where $l_F \ge 0$ is given by (16). Here, $A \succeq B$ means $A - B$ is a positive semidefinite matrix.*

Lemma F.1 is proved in Appendix F.1.

By optimality of $\widehat{\beta}_{ik}$, we have $\mathcal{L}(\widehat{\beta}_{ik}) \leq \mathcal{L}(\beta_i)$ and therefore by (62), we have

$$\frac{1}{2}\langle \widehat{\beta}_{ik} - \beta_i, \nabla^2 \mathcal{L}_{ik}(\tilde{\beta})(\widehat{\beta}_{ik} - \beta_i)\rangle \leq -\langle \nabla \mathcal{L}_{ik}(\beta_i), \widehat{\beta}_{ik} - \beta_i \rangle, \tag{70}$$

where the l.h.s. can be bounded as follows

$$\begin{aligned} \frac{1}{2}\langle \widehat{\beta}_{ik} - \beta_i, \nabla^2 \mathcal{L}_{ik}(\tilde{\beta})(\widehat{\beta}_{ik} - \beta_i)\rangle &= \frac{1}{2}(\widehat{\beta}_{ik} - \beta_i)^{\mathsf{T}} \nabla^2 \mathcal{L}_{ik}(\tilde{\beta})(\widehat{\beta}_{ik} - \beta_i) \\ &\geq \frac{1}{2}(\widehat{\beta}_{ik} - \beta_i)^{\mathsf{T}} l_F \Big(\frac{1}{\ell_{k-1}}\sum_{t \in E_{k-1}} x_t x_t^{\mathsf{T}}\Big)(\widehat{\beta}_{ik} - \beta_i) \\ &= \frac{l_F}{2\ell_{k-1}}(\widehat{\beta}_{ik} - \beta_i)^{\mathsf{T}}\Big(X_k^{\mathsf{T}} X_k\Big)(\widehat{\beta}_{ik} - \beta_i) \\ &= \frac{l_F}{2\ell_{k-1}}\|X_k(\widehat{\beta}_{ik} - \beta_i)\|^2. \end{aligned}$$

Here, $X_k$ is the matrix of size $\ell_{k-1}$ by $d$ whose rows are the feature vectors $x_t$ arriving in episode $k-1$. Moreover, the inequality follows from Lemma F.1. Applying the above bound in Equation (70) and considering the fact that the l.h.s. of this equation is less than or equal to $\|\nabla \mathcal{L}_{ik}(\beta_i)\| \|\widehat{\beta}_{ik} - \beta_i\|$, we get

$$\frac{1}{2\ell_{k-1}} l_F \|X_k(\widehat{\beta}_{ik} - \beta_i)\|^2 \leq \|\nabla \mathcal{L}_{ik}(\beta_i)\| \|\widehat{\beta}_{ik} - \beta_i\|.$$

This implies that on event $\mathcal{E}$, defined in (67), the following holds:

$$\frac{1}{2\ell_{k-1}} l_F \|X_k(\widehat{\beta}_{ik} - \beta_i)\|^2 \leq \lambda_0 \|\widehat{\beta}_{ik} - \beta_i\|. \tag{71}$$

To present a lower bound on the l.h.s. of the above equation, we next lower bound the minimum eigenvalue of $\widehat{\Sigma}_k \equiv (X_k^{\mathsf{T}} X_k)/\ell_{k-1}$. Since rows of $X_k$ are bounded (recall that $\|x_t\| \leq 1$ by our normalization), they are subgaussian. Using [33, Remark 5.40], there exist universal constants $c$ and $C$ such that for every $m \geq 0$, the following holds with probability at least $1 - 2e^{-cm^2}$:

$$\left\|\widehat{\Sigma}_k - \Sigma\right\|_{\mathrm{op}} \leq \max(\delta, \delta^2) \quad \text{where} \quad \delta = C\sqrt{\frac{d}{\ell_{k-1}}} + \frac{m}{\sqrt{\ell_{k-1}}}, \tag{72}$$

where $\Sigma = \mathbb{E}[x_t x_t^{\mathsf{T}}] \in \mathbb{R}^{d \times d}$ is the covariance of the feature vectors. Further, $\|A\|_{\mathrm{op}}$ represents the operator norm of a matrix $A$ and is given by $\|A\|_{\mathrm{op}} = \inf\{c \geq 0 : \|Av\| \leq c\|v\|, \text{ for any vector } v\}$. By our assumption that $\Sigma$ is positive definite[16], we can choose constant $0 < c_{\min} < 1$ such that $\lambda_{\min}(\Sigma) > c_{\min}/d$, with $\lambda_{\min}(A)$ denoting the minimum eigenvalue of a matrix $A$. [17] Set $m = c_{\min}\sqrt{\ell_{k-1}}/(4d)$, $c_0 = (4Cd/c_{\min})^2$ and $c_2 = cc_{\min}^2/16d^2$. Then, for $\ell_{k-1} > c_0 d$ with probability at least $1 - 2e^{-c_2 \ell_{k-1}}$, the following is true:

$$\left\|\widehat{\Sigma}_k - \Sigma\right\|_{\mathrm{op}} \leq \frac{1}{2d}c_{\min}. \tag{73}$$

Denote by $\mathcal{G}$ the probability event that (73) holds. Then, on event $\mathcal{G} \cap \mathcal{E}$, we have

$$\frac{1}{4d}c_{\min} l_F \|\widehat{\beta}_{ik} - \beta_i\|^2 \leq \frac{1}{2\ell_{k-1}} l_F \|X_k(\widehat{\beta}_{ik} - \beta_i)\|^2 \leq \lambda_0 \|\widehat{\beta}_{ik} - \beta_i\|, \tag{74}$$

where the first inequality holds because of Equation (73) and the definition of the operator norm. This results in

$$\|\widehat{\beta}_{ik} - \beta_i\|^2 \leq \frac{16d^2}{c_{\min}^2 l_F{}^2} \lambda_0^2 = \left(\frac{8du_F}{c_{\min} l_F}\right)^2 \left(\sqrt{\frac{\log(\ell_{k-1}d)}{\ell_{k-1}}} + \frac{|\mathcal{L}_{ik}|}{\ell_{k-1}}\right)^2$$

$$\leq 2\left(\frac{8du_F}{c_{\min} l_F}\right)^2 \left(\frac{\log(\ell_{k-1}d)}{\ell_{k-1}} + \left(\frac{|\mathsf{L}_{ik}|}{\ell_{k-1}}\right)^2\right), \tag{75}$$

where in the last line, we used inequality $(a + b)^2 \leq 2a^2 + 2b^2$.

Note that

$$\mathbb{P}((\mathcal{E} \cap \mathcal{G})^c) \leq \mathbb{P}(\mathcal{E}^c) + \mathbb{P}(\mathcal{G}^c) \leq d^{-0.5}\ell_{k-1}^{-1.5} + 2e^{-c_2\ell_{k-1}},$$

and hence the result follows readily from (75), by defining $c_1 \equiv 128(u_F/c_{\min})^2$.

## F.1 Proof of Lemma F.1

We first show the first result in the lemma. We start with few definitions. Let $\tilde{q}_{it} = \mathbb{I}(v_{it} > \max\{b_{-it}^+, r_{it}\})$ be the allocation variable as if buyer $i$ was truthful and the highest competing bid was $b_{-it}^+$. Then, by definition of set of lies $\mathsf{L}_{ik}$, as per (17), for $t \notin \mathsf{L}_{ik}$, we have $q_{it} = \tilde{q}_{it}$. Let

$$\tilde{\mu}_{it}(\beta) = -\tilde{q}_{it}\log'(1 - F(w_{it}(\beta))) - (1 - \tilde{q}_{it})\log'(F(w_{it}(\beta)))$$

be the corresponding quantity to $\mu_{it}(\beta)$, where we replace $q_{it}$ by $\tilde{q}_{it}$. Note that $w_{it}(\beta) = \max\{b_{-it}^+, r_{it}\} - \langle x_t, \beta\rangle$ and $\mu_{it}(\beta)$ is defined in (65). By definition, $\mu_{it}(\beta) = \tilde{\mu}_{it}(\beta)$ for $t \notin \mathsf{L}_{ik}$, and so we can write

$$\nabla \mathcal{L}_{ik}(\beta) = -\frac{1}{\ell_{k-1}}\sum_{t \in E_k}\left(\tilde{\mu}_{it}(\beta)x_t - \mu_{it}(\beta)x_t\right) + \frac{1}{\ell_{k-1}}\sum_{t \in E_{k-1}}\tilde{\mu}_{it}(\beta)x_t$$

$$= -\frac{1}{\ell_{k-1}}\sum_{t \in \mathsf{L}_{ik}}\left(\tilde{\mu}_{it}(\beta)x_t - \mu_{it}(\beta)x_t\right) + \frac{1}{\ell_{k-1}}\sum_{t \in E_{k-1}}\tilde{\mu}_{it}(\beta)x_t.$$

To bound the first term on the right hand side of the last equation, we note that

$$|\mu_{it}(\beta_i)| \leq \sup_{|y| \leq B_n}\left\{\max\left\{\log' F(y), -\log'(1 - F(y))\right\}\right\} = u_F, \tag{76}$$

with $u_F$ given by (68). Here, the first inequality follows from definition of $\mu_{it}(\beta_i)$ as per (65) and using the fact that functions $f$ and $F$ are zero outside the interval $[-B_n, B_n]$. Similarly, we have $|\tilde{\mu}_{it}(\beta_i)| \leq u_F$. Then, considering the fact that $\|x_t\| \leq 1$, we get

$$\left\|\nabla \mathcal{L}_{ik}(\beta_i)\right\| \leq \frac{2u_F}{\ell_{k-1}}|\mathsf{L}_{ik}| + \frac{1}{\ell_{k-1}}\left\|\sum_{t \in E_{k-1}}\tilde{\mu}_{it}(\beta_i)x_t\right\|. \tag{77}$$

We next bound the second term on the right hand side of (77). Define $S_j = \sum_{t=\ell_{k-1}}^{j-1+\ell_{k-1}}\tilde{\mu}_{it}(\beta_i)x_t$, $j = 1, 2, \ldots, \ell_k - 1$, and set $S_0 = 0$. Note that the second term in (77) is equal to $S_{\ell_k-1}$. We upper bound $\frac{1}{\ell_{k-1}}\left\|S_{\ell_k-1}\right\|$ by showing $S_j$ is a vector martingale with bounded differences.

Observe that $\|S_j - S_{j-1}\| \leq u_F\|x_t\| \leq u_F$. Further, $S_j - S_{j-1} = \tilde{\mu}_{it}(\beta_i)x_t$ with $t = \ell_{k-1}+j-1$, and

$$\mathbb{E}[\tilde{\mu}_{it}(\beta_i)|w_{it}(\beta_i)] = \mathbb{P}(\tilde{q}_{it} = 1)\frac{f(w_{it}(\beta_i))}{1 - F(w_{it}(\beta_i))} - \mathbb{P}(\tilde{q}_{it} = 0)\frac{f(w_{it}(\beta_i))}{F(w_{it}(\beta_i))}$$

$$= (1 - F(w_{it}(\beta_i)))\frac{f(w_{it}(\beta_i))}{1 - F(w_{it}(\beta_i))} - F(w_{it}(\beta_i))\frac{f(w_{it}(\beta_i))}{F(w_{it}(\beta_i))} = 0,$$

where the equation holds because $z_{it}$ is independent of $w_{it}(\beta_i)$. Then, considering the fact that $z_{it}$ is independent from the history set (2), we also have

$$\mathbb{E}[S_j - S_{j-1}|S_1, \ldots, S_{j-1}] = \mathbb{E}[\tilde{\mu}_{it}(\beta_i)|S_1, \ldots, S_{j-1}] = 0.$$

So far, we have established that $S_j$ is a matrix martingale with bounded differences. Then, by Matrix Freedman inequality (See Appendix K),

$$\mathbb{P}\Big(\|S_{\ell_{k-1}}\| \geq 2u_F\sqrt{\log(\ell_{k-1}d)\ell_{k-1}}\Big) \leq (d+1)\exp^{-(12/8)\log(\ell_{k-1}d)} = \frac{1}{d^{0.5}\ell_{k-1}^{1.5}}. \quad (78)$$

Then, by Equation (77) and definition of $S_{\ell_{k-1}}$ and event $\mathcal{E}$, given in (67), we have $\mathbb{P}(\mathcal{E}) \geq 1 - d^{-0.5}\ell_{k-1}^{-1.5}$. This completes the proof of the first part of the lemma.

We next prove claim (69) on the Hessian $\nabla^2\mathcal{L}(\beta)$. By characterization (63), it suffices to show that $\eta_{it}(\beta) \geq l_F$. To see this,

$$\begin{aligned}
\eta_{it}(\beta) &= -q_{it}\log''\left(1 - F(w_{it}(\beta))\right) - (1 - q_{it})\log''\left(F(w_{it}(\beta))\right) \\
&\geq \inf_{|y| \leq B_n}\left\{\min\left\{-\log'' F(y), -\log''(1 - F(y))\right\}\right\} \equiv l_F,
\end{aligned}$$

where we used the fact that function $F$ is zero outside the interval $[-B_n, B_n]$. This completes the proof of the lemma.

## G   Proof of Proposition C.2

Here, we need to show claims (19), (20), and (21). Let $o_{it} = (b_{it} - v_{it})_+$ and $s_{it} = (v_{it} - b_{it})_+$ be the amount of overbidding and shading (underbidding) of buyer $i$ in period $t$, respectively. As a common step to show these claims, we upper bound the size of sets $\mathcal{S}_{ik} \equiv \{t : t \in E_{k-1}, q_{it} = 0, s_{it} \geq 1/\ell_{k-1}\}$ and $\mathcal{O}_{ik} \equiv \{t : t \in E_{k-1}, q_{it} = 1, o_{it} \geq 1/\ell_{k-1}\}$. In words, a period $t$ belongs to $\mathcal{S}_{ik}$, if buyer $i$ has shaded his bid significantly in this period, i.e., $s_{it} \geq 1/\ell_{k-1}$, and he does not get the item in this period. Similarly, a period $t$ belongs to $\mathcal{O}_{ik}$, if in this period, buyer $i$ has over-bided by at least $1/\ell_{k-1}$ amount, and he gets the item in this period. We next use the bounds that we establish on $|\mathcal{S}_{ik}|$ and $|\mathcal{O}_{ik}|$ to prove the three aforementioned claims.

To bound the size of sets $\mathcal{S}_{ik}$ and $\mathcal{O}_{ik}$, we use the fact that buyers are utility-maximizer and as a result, they aim for balancing the utility loss due to bidding untruthfully with its potential gain. We define $u_{it}^-$ as the utility that buyer $i$ loses in period $t \in E_{k-1}$ due to bidding untruthfully, relative to the truthful bidding. Precisely, given reserve price $r_{it}$ and the highest competing bid $b_{-it}^+$, $u_{it}^-$ is defined as follows:

$$\begin{aligned}
u_{it}^- &= (v_{it} - \max\{b_{-it}^+, r_{it}\})\,\mathbb{I}(v_{it} > \max\{b_{-it}^+, r_{it}\})\,\mathbb{I}(b_{it} < \max\{b_{-it}^+, r_{it}\}) \\
&\quad - (v_{it} - \max\{b_{-it}^+, r_{it}\})\,\mathbb{I}(v_{it} < \max\{b_{-it}^+, r_{it}\})\,\mathbb{I}(b_{it} > \max\{b_{-it}^+, r_{it}\}).
\end{aligned}$$

Note that the first and second terms are the loss due to underbidding and overbidding, respectively. Our lemma below provides a lower bound on the expected value of $u_{it}^-$.

**Lemma G.1.** *For each buyer $i \in [N]$ and $t \in [\ell_{k-1}, \ell_k - 1]$, we have*

$$\mathbb{E}[u_{it}^-|s_{it}, o_{it}, q_{it}] \geq \frac{1}{2BN\ell_{k-1}}\gamma^t s_{it}^2(1 - q_{it}) + \frac{1}{2BN\ell_{k-1}}\gamma^t o_{it}^2 q_{it}, \quad (79)$$

*where the expectation is taken w.r.t. to the randomness in reserve prices.*

*Proof.* Proof of Lemma G.1. Note that in each period, buyers may suffer a utility loss due to bidding untruthfully. We start with characterizing the impact of underbidding. We then focus on overbidding.

**Underbidding:** After observing the outcome of auction $t$, if buyer $i$ receives the item, then underbidding has no effect on the buyer's instant utility. But if the buyer does not receive the item, then there is a chance that is due to the underbidding. To lower bound $u_{it}^-$, note that in each period $t \in E_{k-1}$, with probability $1/\ell_{k-1}$, the firm does not run a second-price auction. Instead, she picks one of the buyers equally likely and for a reserve price, chosen uniformly at random from $[0, B]$, allocates the item to that buyer if his bid exceeds the corresponding reserve price. Therefore, if a buyer $i$ shades his bid by $s_{it}$, i.e., $s_{it} = (v_{it} - b_{it})_+$, then the utility loss incurred relative to being truthful can be lower bounded as follows:

$$\mathbb{E}[u_{it}^-|s_{it}, q_{it}, v_{it}] \geq \frac{\gamma^t(1 - q_{it})}{BN\ell_{k-1}}\int_{v_{it}-s_{it}}^{v_{it}}(v_{it} - r)\mathrm{d}r = \frac{1}{2BN\ell_{k-1}}\gamma^t s_{it}^2(1 - q_{it}). \quad (80)$$

**Overbidding:** After observing the outcome of auction $t$, if buyer $i$ does not get the item, then overbidding has no effect on the buyer's instant utility. But if the buyer receives the item, then there is a chance that is due to the overbidding. Then, one can follow a similar argument that we used for underbidding to show that

$$\mathbb{E}[u_{it}^-|o_{it}, q_{it}, v_{it}] \ge \frac{\gamma^t(q_{it})}{BN\ell_{k-1}} \int_{v_{it}}^{v_{it}+o_{it}} (-v_{it} + r)\mathrm{d}r = \frac{1}{2BN\ell_{k-1}}\gamma^t o_{it}^2 q_{it}. \tag{81}$$

Then, the result follows from Equations (80) and (81), and by taking expectation w.r.t $v_{it}$, from both sides of these equations, conditioning on $q_{it}$, $s_{it}$, and $o_{it}$.

$$\square \qquad\qquad\qquad\qquad \square$$

With a slight abuse of notation, let $U_{i(k-1)}^-$ be the utility loss of buyer $i$ in episode $k-1$ due to untruthful bidding. That is, $U_{i(k-1)}^- = \sum_{t \in E_{k-1}} u_{it}^-$. In contrast to the utility loss $U_{i(k-1)}^-$, we define $U_{ik}^+$ as the total utility gain that buyer $i$ can achieve by being untruthful in episode $k-1$. More precisely, we fix all other buyer's bidding strategy and consider a reference strategy for buyer $i$. The reference strategy is the same as buyer $i$'s strategy up to episode $k-1$, and in episode $k-1$, the reference policy is just the truthful bidding strategy. We define $U_{ik}^+$ as the total excess utility that buyer $i$ can earn, over the reference strategy. Considering the fact that the bidding strategy of buyer $i$ in episode $k-1$ can only benefit him in the next episodes $k, k+1, \ldots$, we have

$$U_{ik}^+ \le \sum_{t=\ell_k}^{\infty} \gamma^t v_{it} \le B \sum_{t=\ell_k}^{\infty} \gamma^t = B\frac{\gamma^{\ell_k}}{1-\gamma}, \tag{82}$$

where the second inequality holds because $v_{it} \le B$. Indeed, upper bound (82) applies to the total utility any buyer can hope to collect over periods $t \ge \ell_k$.

Now, since we are assuming that the strategic buyers are maximizing their cumulative utility, it must be the case that

$$\mathbb{E}\left[U_{ik}^+ - U_{i(k-1)}^- \,\middle|\, \left\{(q_{it}, s_{it}, o_{it}) : t \in E_{k-1}\right\}\right] \ge 0.$$

Using Lemma G.1 along with upper bound (82), we obtain

$$\sum_{t=\ell_{k-1}}^{\ell_k-1} \frac{1}{2BN\ell_{k-1}}\gamma^t o_{it}^2 q_{it} + \frac{1}{2BN\ell_{k-1}} \sum_{t=\ell_{k-1}}^{\ell_k-1} \gamma^t s_{it}^2(1-q_{it}) \le B\frac{\gamma^{\ell_k}}{1-\gamma}. \tag{83}$$

Next, we lower bound the l.h.s. of the last equation as a function of $|\mathcal{S}_{ik}|$. By definition of $\mathcal{S}_{ik}$, we have

$$\frac{1}{2BN\ell_{k-1}} \sum_{t=\ell_{k-1}}^{\ell_k-1} \gamma^t s_{it}^2(1-q_{it}) \ge \frac{1}{2BN\ell_{k-1}^3} \sum_{t=\ell_k-|\mathcal{S}_{ik}|}^{\ell_k-1} \gamma^t = \frac{\gamma^{\ell_k}}{2BN\ell_{k-1}^3} \cdot \frac{\gamma^{-|\mathcal{S}_{ik}|} - 1}{1-\gamma}.$$

Then, using the last equation along with (83), we get

$$\frac{1}{\ell_{k-1}^3} \cdot \frac{\gamma^{-|\mathcal{S}_{ik}|} - 1}{1-\gamma} \le 2B^2N\frac{1}{1-\gamma},$$

from which we get

$$|\mathcal{S}_{ik}| \le \frac{\log\left(2B^2N\ell_{k-1}^3 + 1\right)}{\log(1/\gamma)} \le C\log(\ell_{k-1}), \tag{84}$$

for a constant $C = C(\gamma, B, N)$. Similarly, one can show that $|\mathcal{O}_{ik}| \le \frac{\log\left(2B^2N\ell_{k-1}^3+1\right)}{\log(1/\gamma)} \le C'\log(\ell_{k-1})$, where $C' = C'(\gamma, B, N)$ is a constant.

So far, we have established an upper bound on $|\mathcal{S}_{ik}|$ and $|\mathcal{O}_{ik}|$. We next bound $|\mathsf{L}_{ik}|$. With this aim, we partition the set of lies $\mathsf{L}_{ik}$ into two subsets $\mathsf{L}_{ik}^s$ and $\mathsf{L}_{ik}^o$, defined below.

$$\mathsf{L}_{ik}^s = \left\{t : t \in E_{k-1}, \mathbb{I}(v_{it} > \max\{b_{-it}^+, r_{it}\}) = 1, \quad \mathbb{I}(b_{it} > \max\{b_{-it}^+, r_{it}\}) = 0\right\}, \tag{85}$$

$$\mathsf{L}_{ik}^o = \left\{t : t \in E_{k-1}, \mathbb{I}(v_{it} > \max\{b_{-it}^+, r_{it}\}) = 0, \quad \mathbb{I}(b_{it} > \max\{b_{-it}^+, r_{it}\}) = 1\right\}. \tag{86}$$

In the following, we bound $|\mathsf{L}_{ik}^s|$ and $|\mathsf{L}_{ik}^o|$ in order to provide an upper bound on $|\mathsf{L}_{ik}|$.

We start with bounding $|\mathsf{L}_{ik}^s|$. Define $\mathcal{S}_{ik}^c \equiv \{t : t \in E_{k-1}, q_{it} = 1 \text{ or } s_{it} < 1/\ell_{k-1}\}$. Then, $|\mathsf{L}_{ik}^s| \leq |\mathcal{S}_{ik}| + |\mathcal{S}_{ik}^c \cap \mathsf{L}_{ik}^s|$. We have already bounded $|\mathcal{S}_{ik}|$. In the following, we bound $|\mathcal{S}_{ik}^c \cap \mathsf{L}_{ik}^s|$.

By definition (85), we first note that for $t \in \mathsf{L}_{ik}^s$, $q_{it} = 0$. Therefore, for $t \in \mathcal{S}_{ik}^c \cap \mathsf{L}_{ik}^s$, we have $s_{it} < 1/\ell_{k-1}$. Let $\mathcal{F}_t \equiv \{(x_\tau, b_{-i\tau}^+, r_{i\tau}) : 1 \leq \tau \leq t\}$. Then, by substituting for $b_{it} = v_{it} - s_{it}$[18] and $v_{it} = \langle x_t, \beta_i \rangle + z_{it}$, we have

$$
\begin{aligned}
&\mathbb{P}(t \in \mathcal{S}_{ik}^c \cap \mathsf{L}_{ik}^s | \mathcal{F}_t) \\
&= \mathbb{P}\left( z_{it} \in [\max\{b_{-it}^+, r_{it}\} - \langle x_t, \beta_i \rangle, \max\{b_{-it}^+, r_{it}\} - \langle x_t, \beta_i \rangle + s_{it}] \text{ and } s_{it} \leq \frac{1}{\ell_{k-1}} \Big| \mathcal{F}_t \right) \\
&\leq \mathbb{P}\left( z_{it} \in \left[ \max\{b_{-it}^+, r_{it}\} - \langle x_t, \beta_i \rangle, \max\{b_{-it}^+, r_{it}\} - \langle x_t, \beta_i \rangle + \frac{1}{\ell_{k-1}} \right] \Big| \mathcal{F}_t \right) \\
&= \int_{\max\{b_{-it}^+, r_{it}\} - \langle x_t, \beta_i \rangle}^{\max\{b_{-it}^+, r_{it}\} - \langle x_t, \beta_i \rangle + 1/\ell_{k-1}} f(z) \mathrm{d}z \leq \frac{c}{\ell_{k-1}}, \quad (87)
\end{aligned}
$$

where the equality holds because $z_{it}$ is independent of $\mathcal{F}_t$. In addition, in the last step, $c \equiv \max_{v \in [-B_n, B_n]} f(v)$ is the bound on the noise density.[19]

Define $\zeta_t \equiv \mathbb{I}(t \in \mathcal{S}_{ik}^c \cap \mathsf{L}_{ik})$ and $\omega_t \equiv \mathbb{P}(t \in \mathcal{S}_{ik}^c \cap \mathsf{L}_{ik} | \mathcal{F}_t)$. Then, $|\mathcal{S}_{ik}^c \cap \mathsf{L}_{ik}| = \sum_{t=\ell_{k-1}}^{\ell_k - 1} \zeta_t$ and $\mathbb{E}(\zeta_t - \omega_t | \mathcal{F}_t) = 0$. Therefore, by using a multiplicative Azuma inequality (see e.g. [23, Lemma 10]), for any $\epsilon \in (0, 1)$ and any $\eta > 0$ we have

$$
\mathbb{P}\left( |\mathcal{S}_{ik}^c \cap \mathsf{L}_{ik}^s| \geq \frac{1+\eta}{1-\epsilon} \sum_{t=\ell_{k-1}}^{\ell_k - 1} \omega_t \right) \leq \exp\left( -\epsilon\eta \sum_{t=\ell_{k-1}}^{\ell_k - 1} \omega_t \right). \quad (88)
$$

We use the shorthand $A \equiv \sum_{t=\ell_{k-1}}^{\ell_k - 1} \omega_t$. By setting $\epsilon = 1/2$, $\eta = (2/A) \log(\ell_{k-1}/\delta)$, the r.h.s of Equation (88) becomes $\delta/\ell_{k-1}$. Further, recalling Equation (87), we have $A \leq \ell_{k-1}(c/\ell_{k-1}) = c$. Hence, rewriting bound (88), we get that with probability at least $1 - \delta/\ell_{k-1}$,

$$
|\mathcal{S}_{ik}^c \cap \mathsf{L}_{ik}^s| = \sum_{t=\ell_{k-1}}^{\ell_k - 1} \zeta_t \leq 2(1+\eta)A \leq 2c + 4\log(\ell_{k-1}/\delta).
$$

Combining the above inequality with bound (84), we get

$$
|\mathsf{L}_{ik}^s| \leq |\mathcal{S}_{ik}^c \cap \mathsf{L}_{ik}^s| + |\mathcal{S}_{ik}| \leq 2c + 4\log(\ell_{k-1}/\delta) + C\log(\ell_{k-1}).
$$

One can establish a similar bound for $|\mathsf{L}_{ik}^o|$. Then, claim (19) follows by using the bounds on $|\mathsf{L}_{ik}^s|$ and $|\mathsf{L}_{ik}^o|$.

To prove claim (20), we write

$$
\begin{aligned}
\sum_{t \in E_{k-1}} s_{it}(1 - q_{it}) &= \sum_{t \in E_{k-1}} s_{it}(1 - q_{it})\mathbb{I}(t \in \mathcal{S}_{ik}) + \sum_{t \in E_{k-1}} s_{it}(1 - q_{it})\mathbb{I}(t \in \mathcal{S}_{ik}^c) \\
&\leq B|\mathcal{S}_{ik}| + \sum_{t \in E_{k-1}} \frac{1}{\ell_{k-1}} \\
&\leq CB\log(\ell_{k-1}) + 1,
\end{aligned}
$$

where we used the fact that $(i)$ $s_{it} \leq v_{it} \leq B$, and $(ii)$ for any $t \in \mathcal{S}_{ik}^c$, either $q_{it} = 1$ or $s_{it} < 1/\ell_{k-1}$. This complete the proof of claim (20).

Finally, we show claim (21). Let $\mathcal{O}_{ik}^c \equiv \{t : t \in E_{k-1}, q_{it} = 0 \text{ or } o_{it} < 1/\ell_{k-1}\}$. Then, we write

$$\sum_{t \in E_{k-1}} o_{it} q_{it} = \sum_{t \in E_{k-1}} o_{it} q_{it} \mathbb{I}(t \in \mathcal{O}_{ik}) + \sum_{t \in E_{k-1}} o_{it} q_{it} \mathbb{I}(t \in \mathcal{O}_{ik}^c)$$

$$\leq M|\mathcal{O}_{ik}| + \sum_{t \in E_{k-1}} \frac{1}{\ell_{k-1}}$$

$$\leq CM \log(\ell_{k-1}) + 1 .$$

Here, we used the fact that $(i)$ $o_{it} \leq M$, and $(ii)$ for any $t \in \mathcal{O}_{ik}^c$, either $q_{it} = 0$ or $o_{it} < 1/\ell_{k-1}$. This completes the proof of claim (21).

## H   Proof of Proposition D.1

The proposition can be proved by following similar steps used in the proof of Proposition C.1. For the quadratic loss function

$$\tilde{\mathcal{L}}_{ik}(\beta) = \frac{1}{|I_k|} \sum_{t \in I_k} (BN q_{it} - \langle x_t, \beta \rangle)^2 ,$$

the gradient and Hessian are given by

$$\nabla \tilde{\mathcal{L}}_{ik}(\beta) = \frac{1}{|I_k|} \sum_{t \in I_k} \mu_{it}(\beta) x_t , \quad \nabla^2 \tilde{\mathcal{L}}_{ik}(\beta) = \frac{1}{|I_k|} \sum_{t \in I_k} 2 x_t x_t^\mathsf{T} , \tag{89}$$

where with a slight abuse of notation, $\mu_{it}(\beta) = 2(\langle x_t, \beta \rangle - BN q_{it})$.

By the second-order Taylor's theorem, expanding around $\beta_i$, we have

$$\tilde{\mathcal{L}}_{ik}(\beta_i) - \tilde{\mathcal{L}}_{ik}(\widehat{\beta}_{ik}) = -\langle \nabla \tilde{\mathcal{L}}_{ik}(\beta_i), \widehat{\beta}_{ik} - \beta_i \rangle - \frac{1}{2} \langle \widehat{\beta}_{ik} - \beta_i, \nabla^2 \tilde{\mathcal{L}}_{ik}(\tilde{\beta})(\widehat{\beta}_{ik} - \beta_{ik}) \rangle , \tag{90}$$

for some $\tilde{\beta}$ on the segment connecting $\beta_i$ and $\widehat{\beta}_{ik}$.

By optimality of $\widehat{\beta}_{ik}$, we have $\tilde{\mathcal{L}}(\widehat{\beta}_{ik}) \leq \tilde{\mathcal{L}}(\beta_i)$ and therefore by (90), we have

$$\frac{1}{2} \langle \widehat{\beta}_{ik} - \beta_i, \nabla^2 \tilde{\mathcal{L}}_{ik}(\tilde{\beta})(\widehat{\beta}_{ik} - \beta_i) \rangle \leq -\langle \nabla \tilde{\mathcal{L}}_{ik}(\beta_i), \widehat{\beta}_{ik} - \beta_i \rangle . \tag{91}$$

Using Equation (89), the r.h.s in the above equation can be written as

$$\frac{1}{2} \langle \widehat{\beta}_{ik} - \beta_i, \nabla^2 \tilde{\mathcal{L}}_{ik}(\tilde{\beta})(\widehat{\beta}_{ik} - \beta_i) \rangle = (\widehat{\beta}_{ik} - \beta_i)^\mathsf{T} \Big( \frac{1}{|I_k|} \sum_{t \in E_{k-1}} x_t x_t^\mathsf{T} \Big) (\widehat{\beta}_{ik} - \beta_i)$$

$$= \frac{1}{|I_k|} (\widehat{\beta}_{ik} - \beta_i)^\mathsf{T} \Big( X_k^\mathsf{T} X_k \Big) (\widehat{\beta}_{ik} - \beta_i)$$

$$= \frac{1}{|I_k|} \| X_k (\widehat{\beta}_{ik} - \beta_i) \|^2 .$$

Here, $X_k$ is the matrix of size $|I_k|$ by $d$, whose rows are the feature vectors $x_t$, with $t \in I_k$ (the exploration phase of episode $k$). Therefore,

$$\frac{1}{|I_k|} \| X_k (\widehat{\beta}_{ik} - \beta_i) \|^2 \leq -\langle \nabla \tilde{\mathcal{L}}_{ik}(\beta_i), \widehat{\beta}_{ik} - \beta_i \rangle \leq \| \nabla \tilde{\mathcal{L}}_{ik}(\beta_i) \| \, \| \widehat{\beta}_{ik} - \beta_i \| . \tag{92}$$

In the next lemma, we bound the gradient of the quadratic loss function. This Lemma is analogous to Lemma F.1.

**Lemma H.1.** *Consider the quadratic loss* (14) *and define the probability event*

$$\mathcal{E} \equiv \big\{ \| \nabla \tilde{\mathcal{L}}_{ik}(\beta_i) \| \leq \lambda_0 \big\}, \quad \text{with} \quad \lambda_0 \equiv 4B(N+1) \sqrt{\frac{\log(\ell_{k-1} d)}{|I_k|}} + 4B(N+1) \frac{|\mathsf{L}_{ik}|}{|I_k|} . \tag{93}$$

*Then, we have* $\mathbb{P}(\mathcal{E}) \geq 1 - d^{-0.5} \ell_{k-1}^{-1.5} .$

Proof of Lemma H.1 is given in Section H.1. Using Lemma H.1 in bound (92), we get

$$\frac{1}{|I_k|}\|X_k(\widehat{\beta}_{ik} - \beta_i)\|^2 \leq \lambda_0 \|\widehat{\beta}_{ik} - \beta_i\| . \tag{94}$$

The proof of Proposition D.1 then follows exactly along the lines after Equation (71) in the proof of its counterpart, Proposition C.1.

## H.1  Proof of Lemma H.1

Let $\tilde{q}_{it} = \mathbb{I}(v_{it} > \max\{b_t^-, r_{it}\})$ be the allocation variables as if buyer $i$ was truthful. Then by definition of set of lies $\mathsf{L}_{ik}$, as per (17), for $t \notin \mathsf{L}_{ik}$, we have $q_{it} = \tilde{q}_{it}$. We define $\tilde{\mu}_{it}(\beta)$ as the counterpart of $\mu_{it}(\beta)$, where we replace $q_{it}$ by $\tilde{q}_{it}$, i.e.,

$$\tilde{\mu}_{it}(\beta) = 2(\langle x_t, \beta \rangle - BN\tilde{q}_{it}) .$$

Recall that $\mu_{it}(\beta) = 2(\langle x_t, \beta \rangle - BNq_{it})$. Since $\mu_{it}(\beta) = \tilde{\mu}_{it}(\beta)$ for $t \notin \mathsf{L}_{ik}$, we can write

$$\begin{aligned}
\nabla \tilde{\mathcal{L}}_{ik}(\beta) &= \frac{1}{|I_k|} \sum_{t \in I_k} \tilde{\mu}_{it}(\beta)x_t - \frac{1}{|I_k|} \sum_{t \in I_k} \{\tilde{\mu}_{it}(\beta)x_t - \mu_{it}(\beta)x_t\} \\
&= \frac{1}{|I_k|} \sum_{t \in I_k} \tilde{\mu}_{it}(\beta)x_t - \frac{1}{|I_k|} \sum_{t \in \mathsf{L}_{ik} \cap I_k} \{\tilde{\mu}_{it}(\beta)x_t - \mu_{it}(\beta)x_t\} .
\end{aligned} \tag{95}$$

To bound $\nabla \tilde{\mathcal{L}}_{ik}(\beta_i)$, we start with bounding $|\mu_{it}(\beta_i)|$ and $|\tilde{\mu}_{it}(\beta_i)|$. By our normalization $\|x_t\| \leq 1$. Further, since $\|\beta_i\| \leq B_p < B$, we obtain $|\langle x_t, \beta_i \rangle| \leq B$. This implies that $|\mu_{it}(\beta_i)| = 2|\langle x_t, \beta_i \rangle - BNq_{it}| \leq 2B(N+1)$. Similarly, we have $|\tilde{\mu}_{it}(\beta_i)| \leq 2B(N+1)$. Therefore, by Equation (95), we have

$$\left\| \nabla \tilde{\mathcal{L}}_{ik}(\beta_i) \right\| \leq \frac{1}{|I_k|} \left\| \sum_{t \in I_k} \tilde{\mu}_{it}(\beta_i)x_t \right\| + \frac{4B(N+1)}{|I_k|}|\mathsf{L}_{ik}| , \tag{96}$$

where we used that $\|x_t\| \leq 1$. To complete the proof of the first part of the lemma, we bound the first term on the right hand side of (96) using the Matrix Freedman inequality for bounded martingale matrices (see Appendix K). Similar to the proof of Lemma F.1, define $S_j = \sum_{t=\ell_k}^{j-1+\ell_k} \tilde{\mu}_{it}(\beta_i)x_t$ and $S_0 = 0$. In order to show that $S_j$ is a vector martingale with bounded differences, we need to show that $\mathbb{E}[\tilde{\mu}_{it}(\beta_i)x_t] = 0$ and bound $\|\tilde{\mu}_{it}(\beta_i)x_t\|$.

Recall that in the pure exploration phase, for a buyer chosen uniformly at random, we set the reserve $r \sim \mathsf{uniform}(0, B)$, and for other buyers we set their reserves to $\infty$. Therefore, for any period $t$ in the pure exploration phase of episode $k$, i.e., for any $t \in I_k$, we have

$$\mathbb{P}(\tilde{q}_{it} = 1|v_{it}, x_t) = \frac{v_{it}}{BN} .$$

As a result, $\mathbb{E}[\tilde{q}_{it}|v_{it}, x_t] = v_{it}/(BN)$, where the expectation is taken w.r.t. to the randomness in reserve prices. Thus,

$$\mathbb{E}[\tilde{\mu}_{it}(\beta_i)|x_t] = 2\mathbb{E}[(BN\mathbb{E}[\tilde{q}_{it}|v_{it}, x_t] - \langle x_t, \beta_i \rangle) \mid x_t] = 2\mathbb{E}[v_{it} - \langle x_t, \beta_i \rangle|x_t] = 2\mathbb{E}[z_{it}|x_t] = 0 . \tag{97}$$

This also implies that $\mathbb{E}[\tilde{\mu}_{it}(\beta_i)x_t] = 0$. Further, $\|\tilde{\mu}_{it}(\beta_i)x_t\| \leq 2B(N+1)\|x_t\| \leq 2B(N+1)$. Thus, by virtue of Matrix Freedman inequality, we have

$$\mathbb{P}\left(\frac{1}{|I_k|} \left\| \sum_{t \in I_k} \tilde{\mu}_{it}(\beta_i)x_t \right\| \geq 4B(N+1)\sqrt{\frac{\log(\ell_{k-1}d)}{|I_k|}}\right) \leq (d+1)\exp^{-(12/8)\log(\ell_{k-1}d)} = \frac{1}{d^{0.5}\ell_{k-1}^{1.5}} . \tag{98}$$

Combining Equations (96) and (98) shows that $\mathbb{P}(\mathcal{E}) \geq 1 - d^{-0.5}\ell_{k-1}^{-1.5}$, where the probability event $\mathcal{E}$ is defined in (93).

# I Proof of Proposition D.2

The proof is based on comparing the utility loss of an untruthful buyer with his future utility gain and using the fact that buyers are utility-maximizing. Note that in SCORP, any utility loss due to untruthful bidding can only happen in the exploration phase of the episodes, as the submitted bids in the exploitation phase of the episodes are not used in estimating the preference vectors. Therefore, during the exploitation phase, there is no incentive for buyers to deviate from being truthful. Hence, to bound the utility loss of a buyer due to untruthful bidding, we only need to focus on the pure exploration phase.

By focusing on the exploration phase, it is easy to verify that by following similar steps as in the proof of Lemma G.1, we have

$$\mathbb{E}[u_{it}^- | s_{it}, o_{it}, q_{it}] \geq \frac{1}{2BN} \gamma^t s_{it}^2 (1 - q_{it}) + \frac{1}{2BN} \gamma^t o_{it}^2 q_{it}, \tag{99}$$

where the expectation is taken w.r.t. to the randomness in reserve prices.

Observe that this bound is stronger than Lemma G.1 in that the factor $1/(2BN\ell_{k-1})$ is replaced by $1/2BN$. The reason is that *in each period* of pure exploration phase, for a randomly chosen buyer we set his reserve $r \sim \mathsf{uniform}(0, B)$ and we set other buyer's reserves to $\infty$. This is in contrast to the CORP policy (under known distribution $F$) that we do such exploration only with probability $1/\ell_{k-1}$ in each period of episode $k$. We remove the proof of Equation (99), as it is very similar to the proof of Lemma G.1.

By having Equation (99) in place, the rest of the proof is exactly the same as the proof of Proposition C.2.

# J Proof of Technical Lemmas

## J.1 Proof of Lemma C.3

Note that in a second-price auction with truthful buyers, $W_{it}(r)$ indicates the revenue that firm earns when buyer $i$ wins the auction and has been posted reserve price $r$. Therefore by definition of optimality $r_{it}^\star = \arg\max_r W_{it}(r)$. (In Proposition 3.2, it is shown that $r_{it}^\star$ is the optimal solution of optimization problem (4).) Therefore, $W_{it}'(r_{it}^\star) = 0$.

Also, by Equation (60), we have

$$W_{it}'(r) = H_{it}(r)\Big((1 - F(r - \langle x_t, \beta_i \rangle)) - rf(r - \langle x_t, \beta_i \rangle)\Big). \tag{100}$$

Hence,

$$\begin{aligned} W_{it}''(r) = {} & h_{it}(r)\Big((1 - F(r - \langle x_t, \beta_i \rangle)) - rf(r - \langle x_t, \beta_i \rangle)\Big) \\ & - 2H_{it}(r)f(r - \langle x_t, \beta_i \rangle) - H_{it}(r)rf'(r - \langle x_t, \beta_i \rangle). \end{aligned} \tag{101}$$

Since valuations and bids are bounded by constant $B$, clearly $0 \leq r_{it}^\star, r_{it} \leq B$ and given that $r$ is between them, we also have $0 \leq r \leq B$. In addition, considering the fact that the market noise is bounded in $[-B_n, B_n]$ and $f$ and $f'$ are continuous, both $f$ and $f'$ attain their maximum over the compact interval $[-B_n, B_n]$. Let $c_1 = \max_{y \in [-B_n, B_n]} f(y)$ and $c_2 = \max_{y \in [-B_n, B_n]} f'(y)$. Further, since $0 \leq v_t^- \leq B$, its density $h_{it}$ is supported in $[-B, B]$ and due to continuity, it attains its maximum over this interval. Let $c_3 = \max_{y \in [-B, B]} h(y)$. Therefore,

$$|W_{it}''(r)| \leq c_3 + 2c_1 + Bc_2.$$

The result follows by setting $c \equiv c_3 + 2c_1 + Bc_2$.

## J.2 Proof of Lemma C.4

We define function $g : \mathbb{R} \mapsto \mathbb{R}$ as follows:

$$g(\theta) = \arg\max_y \{y(1 - F(y - \theta))\}. \tag{102}$$

By this definition, for any $t \in E_k$, we have $r_{it}^\star = g(\langle x_t, \beta_i \rangle)$ and $r_{it} = g(\langle x_t, \widehat{\beta}_{ik} \rangle)$. Then, by showing $g(\cdot)$ is 1-Lipschitz function, claim (52) follows. To see this note that

$$|r_{it}^\star - r_{it}| = |g(\langle x_t, \beta_i \rangle) - g(\langle x_t, \widehat{\beta}_{ik} \rangle)| \leq |\langle x_t, \beta_i - \widehat{\beta}_{ik} \rangle|, \tag{103}$$

where the inequality holds because of 1-Lipschitz property of function $g$.

By definition (102), $g(\theta)$ should satisfy the following stationary condition:

$$1 - F(g(\theta) - \theta) = g(\theta)f(g(\theta) - \theta).$$

Define $\varphi(y) \equiv y - \frac{1-F(y)}{f(y)}$ as the *virtual valuation* function. Then, we can write $g(\theta)$ in terms of virtual valuation function: $\varphi(g(\theta) - \theta) = -\theta$. Since $\varphi$ is injective, by applying $\varphi^{-1}$ to both sides, we can write $g(\theta)$ explicitly in terms of virtual valuation function:

$$g(\theta) = \theta + \varphi^{-1}(-\theta). \tag{104}$$

Using characterization (104), we show that $g$ is 1-Lipschitz. To do so, we verify $g'(\theta) = 1 - 1/\varphi'(\varphi^{-1}(-\theta))$ is less than one. In particular, $g'(\theta) \leq 1$ because $\varphi'(y) \geq 1$. To see why this holds note that $\varphi(y)$ can be written as $\varphi(y) = y + \frac{1}{\log'(1-F(y))}$. Then, by Assumption 3.1, $1 - F$ is log-concave. This implies that $\log'(1 - F(y))$ is decreasing, and consequently $\varphi(y)$ is increasing. Indeed, this implies that $\varphi'(y) \geq 1$.

## J.3  Proof of Lemma C.5

We first prove Claim (34). Observe that when $v_t^- - b_t^- < 0$, Claim (34) holds, as $s_{it} \geq 0$ for any $i \in [N]$. Thus, it suffices to show that $(v_t^- - b_t^-) \leq \max\{s_{it}(1 - q_{it}) : i \in [N]\}$. Without loss of generality, assume $v_{1t} > v_{2t} > \ldots > v_{Nt}$. Then, $v_t^- = v_{2t}$. If $b_t^- \geq b_{2t}$, then buyer 2 will not receive the item, i.e., $q_{2t} = 0$ and we have

$$v_t^- - b_t^- = v_{2t} - b_t^- \leq v_{2t} - b_{2t} = (v_{2t} - b_{2t})(1 - q_{2t}) = s_{2t}(1 - q_{2t}),$$

proving the claim in this case. The other case is when $b_t^- < b_{2t}$ and hence $b_{2t}$ is the highest bid. This implies that $b_{1t} \leq b_t^-$ and we have the following chain of inequalities:

$$v_t^- - b_t^- = v_{2t} - b_t^- \leq v_{2t} - b_{1t} < v_{1t} - b_{1t}. \tag{105}$$

Further, since buyer 2 has the highest bid, $q_{2t} = 1$ and $q_{it} = 0$ for all $i \neq 2$. In particular, $q_{1t} = 0$. Combining this with (105), we get

$$v_t^- - b_t^- < (v_{1t} - b_{1t})(1 - q_{1t}),$$

which proves the claim in this case as well.

We next prove Claim (35). Suppose $q_{it} = 0$ and let buyer $j$ be the winner ($q_{jt} = 1$ and $j \neq i$). Then, by definition $b_{-it}^+ = b_{jt}$ and

$$b_{-it}^+ - v_{-it}^+ = b_{jt} - v_{-it}^+ \leq b_{jt} - v_{jt} = o_{jt}q_{jt}.$$

Here, we use that $v_{jt} \leq v_{-it}^+$ because $j \neq i$.

## J.4  Proof of Lemma D.3

Recall that $r_{it}^\star$ and $r_{it}$ are given by the following equations:

$$r_{it}^\star = \arg\max_r \min_{F \in \mathcal{F}} r(1 - F(r - \langle x_t, \beta_i \rangle)),$$

$$r_{it} = \arg\max_r \min_{F \in \mathcal{F}} r(1 - F(r - \langle x_t, \widehat{\beta}_{ik} \rangle)).$$

Let $\tilde{r}_{it}^\star = r_{it}^\star - \langle x_t, \beta_i \rangle$ and $\tilde{r}_{it} = r_{it} - \langle x_t, \widehat{\beta}_{ik} \rangle$. By a change of variable, it is easy to see that $\tilde{r}_{it}^\star$ and $\tilde{r}_{it}$ are the solutions to the following optimization problems:

$$\tilde{r}_{it}^\star = \arg\max_r \min_{F \in \mathcal{F}} \left\{ (r + \langle x_t, \beta_i \rangle)(1 - F(r)) \right\},$$

$$\tilde{r}_{it} = \arg\max_r \min_{F \in \mathcal{F}} \left\{ (r + \langle x_t, \widehat{\beta}_{ik} \rangle)(1 - F(r)) \right\}.$$

Define function $H : \mathbb{R} \to \mathbb{R}$ as $H(r) \equiv \max_{F \in \mathcal{F}} F(r)$. Observe that $\tilde{r}_{it}^\star + \langle x_t, \beta_i \rangle = r_{it}^\star > 0$ and hence,

$$\begin{aligned}
\tilde{r}_{it}^\star &= \arg\max_r \ \min_{F \in \mathcal{F}} \ (r + \langle x_t, \beta_i \rangle)(1 - F(r)) \\
&= \arg\max_r \ (r + \langle x_t, \beta_i \rangle) \min_{F \in \mathcal{F}}(1 - F(r)) \\
&= \arg\max_r \ (r + \langle x_t, \beta_i \rangle)(1 - H(r)) \,.
\end{aligned} \tag{106}$$

Using the change of variable $r \leftarrow r + \langle x_t, \beta_i \rangle$, we obtain

$$r_{it}^\star = \arg\max_r \ r(1 - H(r - \langle x_t, \beta_i \rangle)) \,. \tag{107}$$

Likewise,

$$r_{it} = \arg\max_r \ r(1 - H(r - \langle x_t, \widehat{\beta}_{ik} \rangle)) \,. \tag{108}$$

Now, note that by definition of function $H$, we have $\log(1 - H(r)) = \min_{F \in \mathcal{F}} \log(1 - F(r))$. Further, $F$ is log-concave for all $F \in \mathcal{F}$, as per Assumption B.1. Moreover, using the fact that the (pointwise) minimum of concave functions is also concave, we have that $1 - H$ is log-concave. By virtue of characterizations (107) and (108), and log-concavity of $1 - H$, the claim follows from the same proof of Lemma C.4 and hence is omitted. The only subtle point is that function $H$, although continuous, may not be differential at some points. Therefore, in using the argument of Lemma C.4, derivative should be replaced by subgradient.

## K   Matrix Freedman Inequality

For readers' convenience, here we state the Matrix Freedman inequality for martingales.

**Theorem K.1** (Rectangular Matrix Freedman). *Consider a matrix martingale $\{Y_k : k = 0, 1, 2, \ldots\}$ whose values are matrices with dimension $d_1 \times d_2$ and let $\{X_k : k = 1, 2, \ldots\}$ be the difference sequence. Assume that the difference sequence is uniformly bounded:*

$$\|X_k\|_{\mathrm{op}} \le R \quad \text{almost surely} \quad \text{for } k \ge 1 \,, \tag{109}$$

*where $\| \cdot \|_{\mathrm{op}}$ denotes the operator norm[20], and $R$ is a constant. Define two predictable quadratic variation processes for this martingale*

$$W_{1,k} \equiv \sum_{j=1}^{k} \mathbb{E}[X_j X_j^\mathsf{T} | Y_1, \ldots, Y_{j-1}] \,,$$

$$W_{2,k} \equiv \sum_{j=1}^{k} \mathbb{E}[X_j^\mathsf{T} X_j | Y_1, \ldots, Y_{j-1}] \,,$$

*for $k \ge 1$. Further, for given $\sigma^2 > 0$ and $t \ge 0$, let event $\mathcal{A} \equiv \big\{ \exists k \ge 0 : \ \|Y_k\|_{\mathrm{op}} \ge t, \ \max(\|W_{1,k}\|_{\mathrm{op}}, \|W_{2,k}\|_{\mathrm{op}}) \le \sigma^2 \big\}$. Then,*

$$\mathbb{P}(\mathcal{A}) \le (d_1 + d_2) \exp\left(-\frac{t^2/2}{\sigma^2 + RT/3}\right) = \begin{cases} (d_1 + d_2) \exp(-3t^2/8\sigma^2) & t \le \sigma^2/R \,, \\ (d_1 + d_2) \exp(-3t/8R) & t \ge \sigma^2/R \,. \end{cases}$$

We refer to [32] for the proof of Theorem K.1. We next state the result of Matrix Freedman theorem specialized to the vector case. This corollary is used in the proof of Propositions F.1 and H.1.

**Corollary K.2.** *Consider a vector martingale $\{u_k : k = 0, 1, 2, \ldots\}$ whose values are vector with dimension $d$ and let $\{v_k : k = 1, 2, \ldots\}$ be the difference sequence. Assume that the difference sequence is uniformly bounded:*

$$\|v_k\| \le R \quad \text{almost surely} \quad \text{for } k \ge 1 \,, \tag{110}$$

*Define a predictable quadratic variation processes for this martingale:*

$$w_k \;\equiv\; \sum_{j=1}^{k} \mathbb{E}\Big[\|v_j\|^2 \Big| u_1, \dots, u_{j-1}\Big], \quad \text{for } k \geq 1.$$

*Then, for all $t \geq 0$ and $\sigma^2 > 0$, we have*

$$\mathbb{P}\Big\{\exists k \geq 0 : \|u_k\| \geq t \quad \text{and} \quad \|w_k\| \leq \sigma^2\Big\} \;\leq\; (d+1)\exp\left(-\frac{t^2/2}{\sigma^2 + Rt/3}\right) \tag{111}$$

$$= \begin{cases} (d+1)\exp(-3t^2/8\sigma^2) & \text{for } t \leq \sigma^2/R, \\ (d+1)\exp(-3t/8R) & \text{for } t \geq \sigma^2/R. \end{cases}$$

We used Corollary K.2 in the proof of Lemma F.1 to bound the norm of martingale $S_j$ (see below Equation (77)). Specifically, we used the corollary with $u_j = S_j$, $v_j = \tilde{\mu}_{it}(\beta_i)x_t$ (with $t = \ell_{k-1}+j-1$), $R = u_F$, and $\sigma^2 = u_F^2 \ell_{k-1}$. Then, using bound (111) for $S_{\ell_{k-1}}$ with $t = 2u_F\sqrt{\log(\ell_{k-1}d)\ell_{k-1}}$, we obtain bound (78).

Likewise, we used Corollary K.2 in the proof of Lemma H.1 to bound the norm of martingale $S_j$ (see below Equation (96)). Here, again we set $u_j = S_j$, $v_j = \tilde{\mu}_{it}(\beta_i)x_t$ with $t = \ell_{k-1}+j-1$ (note that in this case $\tilde{\mu}_{it}(\beta_i) = 2(\langle x_t, \beta_i \rangle - BN\tilde{q}_{it})$.) We then have $R = 2B(N+1)$ and $\sigma^2 = 4B^2(N+1)^2 j$. We then use bound (111) for $S_{\ell_{k-1}}$ with $t = 4B(N+1)\sqrt{\log(\ell_{k-1}d)|I_k|}$ to obtain (98).

## Footnotes

[6]The term $BN$ in $\tilde{\mathcal{L}}_{ik}(\beta)$ is a normalization factor and is due to the fact that in the exploration phase, one of the $N$ buyers are chosen uniformly at random and the chosen buyer is offered the item at price of $r \sim \mathsf{uniform}(0, B)$.

[7]The firm may care about other objectives apart from her revenue. For instance, the firm might be interested in maximizing a convex combination of the welfare and revenue, or due to contracts and deals, she might be willing to prioritize some of the buyers by offering them lower reserve prices.

[8]The constants $c_3$ and $c_4$ depend on $\gamma$, $B$, and $N$. Constant $c_5$ depends on $\gamma$, $M$, and $N$. (Recall that $M$ is the bound on submitted bids)

[9]This follows from the Remainder theorem for Taylor's expansion.

[10]Note that by our normalization, the sum of eigenvalues of $\Sigma$ would be $\text{trace}(\Sigma) = \mathbb{E}[\|x_t\|^2] \leq 1$, and that is why the eigenvalues are scaled by $1/d$.

[11]Note that the density $f$ is continuous and hence attains its maximum over compact sets.

[12]Indeed, we can make truthfulness the unique best response strategy in the exploitation phase by tweaking the mechanism such that with a fixed small probability in each round, all the reserve prices are set to zero, independently.

[13]Note that at a negative value of $y$, function $y \mapsto y(1 - F(y - \langle x_t, \beta_i \rangle))$ is also negative and hence this function cannot take its maximum at a negative $y$.

[14]If $h(y)$ is strictly log-concave function, then at a stationary point $y_0$ that $h'(y_0) = 0$, we have $\log'(h(y_0)) = h'(y_0)/h(y_0) = 0$. Given that $\log(h(y))$ is strictly concave, this means that $y_0$ is the unique global maximizer of $\log(h(y))$ and by strict monotonicity of the logarithm function, this implies that $y_0$ is also the unique global maximizer of $h(y)$.

[15]Since the density $f$ is zero outside the interval $[-B_n, B_n]$, we have $F(z) = 0$ for $z < -B_n$, and $F(z = 1)$ for $z > B_n$. In Equation (64), if $w_{it}(\beta)$ is outside the interval $[-B_n, B_n]$, we use the convention of $\frac{0}{0} = 0$.

[16]A symmetric matrix is said to be positive definite if all of its eigenvalues are strictly positive. In general, if the distribution of features $\mathcal{D}$, is bounded below from zero on an open set around the origin, then its second-moment matrix is positive definite. This assumption holds for many common distributions such as normal and uniform distributions.

[17]Note that by our normalization, the sum of eigenvalues of $\Sigma$ would be $\mathrm{trace}(\Sigma) = \mathbb{E}[\|x_t\|^2] \leq 1$, and that is why the eigenvalues are scaled by $1/d$.

[18]Note that here $s_{it} > 0$ as $t \in \mathsf{L}_{ik}^s$ and consequently $b_{it} < v_{it}$.

[19]Note that the density $f$ is continuous and hence attains its maximum over compact sets.

[20]For a matrix $A$, its operator norm is defined as $\|A\|_{\mathrm{op}} = \inf\{c \ge 0 : \|Av\| \le c\|v\|, \text{ for any vector } v\}$. Equivalently, the operator norm is the largest singular value of a matrix.