[Reviews · NeurIPS 2019]

Reviewer 1



The authors study the problem of setting (individual) reserve prices in a scenario of repeated contextual second-price auctions. The buyers are assumed strategic, i.e. they optimize a cumulative discounted utility, where their valuations are linear functions of the feature vector of a good. The considered scenario explicitly assumes existence of noise in the market. The seller’s goal is to find an algorithm for setting prices that has sub-linear regret. Two algorithms are proposed: - the first one attain O(d log(Td) log(T)) regret bound, when the market noise distribution is known to the seller. - the other one attain O(d (log (Td))^1/2 T^2/3) regret bound, when the market noise distribution is unknown to the seller. T is the horizon of the game, while d is the dimension of the feature vector space. The authors conducted a large study. The results are technically good. However, the presentation of the work can be significantly improved. Key weaknesses: 1. Positioning with respect to related work: - The authors compare their work with literature on the non-contextual single strategic buyer case and cite [1,25]. However, for this case, the state-of-the-art solutions have been found recently, see (Drutsa, WWW’2017) and (Drutsa, ICML’2018), where first algorithms with optimal regret bound of O(log log T) are presented. It seems that the authors are unaware of those studies. Thus, it is unclear in the paper, whether the proposed algorithms (e.g., CORP) are optimal and is it possible to find an algorithm with a more favorable regret bound (since, in the non-contextual case, the bound is doubly logarithmic). - The buyer valuation is a linear function of a good’s features. This approach assumes that the seller knows in advance the type of the model used by the buyer, what seems too unrealistic. There are works on multi-dimensional non-parametric search, e.g., (Mao et al., NeurIPS’2018) - Besides [2], there is a lot of works on contextual pricing for auctions, where buyers are not strategic. For instance, [11, 18] and (Leme and Schneider, FOCS’2018). I hope a light on this can be done in the paper. - Moreover, Cohen et al. in [11] also provided tricks for robustness of their algorithm (including a stochastic noise). Could you position your ways to fight noise w.r.t. that study, please? [After author feedback phase: Thank you for the clear discussion of the related studies. Please, include these discussions in the final version of your paper. Such a discussion of related work will help ML community to better understand the position and the scope of the study.] 2. The study deals with assumption that the buyer valuation linearly depends on the features of a good. What if the dependence is not linear? [After author feedback phase: Great, it is a good way to extend your result. I hope you add it in the next version of your paper] 3. There is no conclusion of the study. Refs: Drutsa, “Horizon-independent optimal pricing in repeated auctions with truthful and strategic buyers”, WWW’2017: Drutsa, “Weakly Consistent Optimal Pricing Algorithms in Repeated Posted-Price Auctions with Strategic Buyer”, ICML’2018 Mao, Leme, Schneider, “Contextual pricing for Lipschitz buyers”, NeurIPS’2018 Leme and Schneider, "Contextual search via intrinsic volumes", FOCS'2018.

Reviewer 2



This work considers a revenue maximizing seller trying to set personalized reserve prices for a contextual repeated single-item auction among a number of strategic bidders. Importantly, the bidders must have a lower time discount factor than the seller, as this is used to balance the incentive to influence learning from the buyers. The paper proposes two pricing schemes: CORP and SCORP, for the cases that there is known and unknown noise in the bidders value functions respectively. Both schemes are explore/exploit schemes that balance the number of periods of explore vs exploit to curtail the revenue impact from misreports as well as the number of periods in which there could be an incentive to misreport. Importantly, in the explore phase the auction presents one random bidder with a random take it or leave it price and only leverages whether or not the item was purchased for its learning of valuations. This is critical to the analysis as it ensures that the utility penalty due to over or underreporting is high relative to the amount to gain from misreporting in the future. Significance - This work is really the first to tackle learning contextual reserve pricing with multiple strategic bidders and it's approaches should be useful for future research in the space. [Added in discussion period] The paper is very well written, particularly for each pricing policy. The core proofs are very involved - I followed the sketches, but am not able to verify the full proofs. Minor notes: - 327, 468: COPR - 842: over-bided

Reviewer 3



Overall, I found this paper a joy to read. The problem is explained clearly, adequate context is given in the related work section, and the main paper focuses on assumptions, intuition and take-aways rather than getting bogged down in techincal details. I'm not a world-expert on learning with auctions, and rather partially aware of the literature, but that did not hinder me. Given the wide range of interests at NeurIPS, I think that this paper is of interests to those for whom auctions and strategic behavior is not their primary focus. I have a few minor comments: - I'm not particularly convinced by the motivation for buyers using a discount factor, i.e. being less patient than the firm (Paragraph starting at 248). For example, in the targeting example it seems to me that the user who shows up today is not the same as the one showing up tomorrow. Whether a user just visited the site of the buyer seems better modeled by the context rather than by a discount factor. Similarly, I feel that the context should capture the value of using cloud computing capacity rather than a discount factor. I think one can reasonably defend the modeling assumption, and I think the paper could do a better job. - I like the discussion of important features (line 289 and beyond) a lot, and would suggest adding a heading to guide readers to this content; currently, it's easy to overlook. - Given that the discount factor plays such a big role in the results of the paper, it would be nice to have the results explicitly depend on the discount factor (Theorem 4.1). - Footnote 5 raises an interesting question whether a firm can do better by committing to some strategy that does not explicitly try to maximize it's own revenue (for example, by attracting more buyers to the platform). Stated less paradoxically; how does the strategy of the firm relate to the number of buyers on the platform, rather than the number of buyers considered fixed. --- Post rebuttal --- Overall, I think the authors addressed concerns raised in initial review well and incorporating their response into the paper seems beneficial. I'm not fully convinced yet by the justification for the discount factor though (to be clear, I don't mind the assumption but just think the justification is a bit shaky). E.g. in cloud computing the authors write that opportunity costs exists because many tasks are time-sensitive. That makes sense if the same task persists over time (I rather get the task done now than in the next time period). But tasks are different each round, and so there does not seem to be a way to capture time sensitivity. All in all, great job and hope to see the paper accepted!

[Author Response · NeurIPS 2019]

We thank the reviewers for their insightful comments and suggestions on our paper.

**Reviewer 1:** Thanks for pointing out these related papers. We discuss them below and plan that upon acceptance use
the extra page to add these discussions to the paper as well.

• [Drusta WWW2017, ICML 2018]: These work focus on single buyer and a single product that is offered repeatedly.
The (private) buyer's valuation of this product remains fixed across time. For case of strategic buyer, a discounted regret
is considered with factor $\gamma < 1$ (similar to ours). An FES and PRRFES pricing policies are proposed which achieve
the regret $O(\log \log T)$. However, their setting is very different than ours in that we consider 1) multi-buyer 2) feature
based setting and the products offered at different rounds could be highly different. As a result, the buyers' valuations
(that depend on the product features and also market noise) vary over time. For this we do not expect to have a policy
with $O(\log \log T)$ regret in our setting. Closely related, Broder& Rusmevichientong (2012) consider the setting of a
single truthful buyer and a retailer (single product) under a parametric choice model (corresponding to noisy valuation)
and prove a lower bound of $O(\log T)$ for regret, using Van Trees inequality. One can follow that approach and via
generalized Van Trees inequality prove a lower bound $O(d \log T)$ for the setting of CORP policy.

• Extension to nonlinear models: Although the paper focuses on linear valuation models, it is straightforward to
generalize our analysis to some of the nonlinear valuation models. Specifically, consider model

$$v_{it}(x_t) = \psi(\langle \phi(x_t), \beta_i \rangle + z_{it}) \quad i \in [N], \quad t \geq 1,$$

where $\phi : \mathbb{R}^d \mapsto \mathbb{R}^d$ is a mapping and $\psi : \mathbb{R} \mapsto \mathbb{R}$ is an increasing function. Then by the change of variable
$\tilde{v}_{it} = \psi^{-1}(v_{it})$, $\tilde{x}_t = \phi(x_t)$ , we arrive at the relation $\tilde{v}_{it} = \langle \tilde{x}_t, \beta_i \rangle + z_{it}$. By modifying the CORP policy for this
relation, we can get a policy that also achieve logarithmic regret for these nonlinear settings. Some examples include:
log-log model, semi-log model and logistic model. While these models have been popular for some applications (the
first two in hedonic pricing and the last in click-through-rate prediction), it is still an interesting direction to consider
nonparametric models (e.g., Mao et. al. 2018 , Chen & Gallego 2018) but it is beyond the scope of current paper.

• With respect to Cohen et al. (and their tricks for robustness), their modified policy gets a regret of
$O(d^2 \log(\min\{T/d, 1/\delta\}) + d\delta T)$, where $\delta$ measures the noise magnitude: in case of bounded noise, $\delta$ represents the
uniform bound on noise and in case of gaussian noise with variance $\sigma^2$, it is defined as $\delta = 2\sigma\sqrt{\log(T)}$. But then to
have a logarithmic regret, $\delta$ should scale $O(\frac{\log T}{T})$, which we find very restrictive (not clear why the noise should shrink
with time horizon and at a rate $1/T$). In comparison, we do not require such assumption.

• We will add "Conclusion" section in the revision and in our Related work section, we will add the following w.r.t
contextual dynamic pricing with learning in non-strategic environment: Chen et al. (2015) studied this problem when
the demand function follows the logit model and proposed an ML-based learning algorithm. Leme and Schneider
(2018), Cohen et al. (2016), and Lobel et al. (2016) proposed a learning algorithm based on the binary search method
when the demand function is linear and deterministic. In their models, buyers have homogenous preference vectors
and are non-strategic. Hence, the problem reduces to a single buyer setting, where the buyer acts myopically, i.e., the
buyer does not consider the impact of the current actions on the future prices. There is also a new line of literature that
studied dynamic pricing with demand learning when the contextual information is high dimensional (but sparse); see
Javanmard and Nazerzadeh (2019), Ban and Keskin (2017).

**Reviewer 2:** Thanks for your positive comments. We will fix the minor typos in the revision. For the lower bound
please see the response to Reviewer 1.

**Reviewer 3: (Discount factor:)** To make the dependence of regret on the discount factor explicit, the regret bound
(Theorem 4.1) works out at $O(\log(Td) \log(T) + \frac{\log^2(T)}{\log^2(1/\gamma)})$. Notably, the first term is due to the estimation error in
preference vectors; that is, this term exists even if buyers were not strategic. The second term is due to the strategic
behavior of the buyers and decreases as buyers get less patient ($\gamma$ gets smaller).

Regarding the justification for discount factor: two types of targeting are common in online advertising (using HTTP
cookies and using demographic information). While the former can potentially identify a unique user, the latter targets a
group of users. We agree that for targeting based HTTP cookies, one can use time-varying contextual models. However,
for targeting based on demographic information, using discount factors is more suitable. Specially that the number
of users from these demographics is uncertain and the advertiser would like to keep showing the ads not to miss the
opportunity of displaying his ad to a right user. The discount factor aims at capturing this opportunity cost. In cloud
computing, a similar opportunity cost exists because many tasks are time sensitive.

• Footnote 5: We believe that one can extend our algorithms to a setting in which the seller has different objective
functions such as weighted sum of buyers' welfare and his revenue or he aims at maximizing his revenue subject to
return-on-investment (ROI) constraints. The difference is that the "optimal price" should be lower.

[Meta-Review · NeurIPS 2019]

Reviewers expressed some concerns in their initial reviews, but are satisfied with the author response. The consensus is to "accept".